# On Regret with Multiple Best Arms

**Yinglun Zhu**
Department of Computer Sciences
University of Wisconsin-Madison
Madison, WI 53706
yinglun@cs.wisc.edu

**Robert Nowak**
Department of Electrical and Computer Engineering
University of Wisconsin-Madison
Madison, WI 53706
rdnowak@wisc.edu

## Abstract

We study a regret minimization problem with the existence of multiple best/near-optimal arms in the multi-armed bandit setting. We consider the case when the number of arms/actions is comparable or much larger than the time horizon, and make *no* assumptions about the structure of the bandit instance. Our goal is to design algorithms that can automatically adapt to the *unknown* hardness of the problem, i.e., the number of best arms. Our setting captures many modern applications of bandit algorithms where the action space is enormous and the information about the underlying instance/structure is unavailable. We first propose an adaptive algorithm that is agnostic to the hardness level and theoretically derive its regret bound. We then prove a lower bound for our problem setting, which indicates: (1) no algorithm can be minimax optimal simultaneously over all hardness levels; and (2) our algorithm achieves a rate function that is Pareto optimal. With additional knowledge of the expected reward of the best arm, we propose another adaptive algorithm that is minimax optimal, up to polylog factors, over *all* hardness levels. Experimental results confirm our theoretical guarantees and show advantages of our algorithms over the previous state-of-the-art.

## 1 Introduction

Multi-armed bandit problems describe exploration-exploitation trade-offs in sequential decision making. Most existing bandit algorithms tend to provide regret guarantees when the number of available arms/actions is smaller than the time horizon. In modern applications of bandit algorithm, however, the action space is usually comparable or even much larger than the allowed time horizon so that many existing bandit algorithms cannot even complete their initial exploration phases. Consider a problem of personalized recommendations, for example. For most users, the total number of movies, or even the amount of sub-categories, far exceeds the number of times they visit a recommendation site. Similarly, the enormous amount of user-generated content on YouTube and Twitter makes it increasingly challenging to make optimal recommendations. The tension between a very large action space and a limited time horizon poses a realistic problem in which deploying algorithms that converge to an optimal solution over an asymptotically long time horizon *do not* give satisfying results. There is a need to design algorithms that can exploit the highest possible reward within a *limited* time horizon. Past work has partially addressed this challenge. The quantile regret proposed in [12] to calculate regret with respect to an satisfactory action rather than the best one. The discounted regret analyzed in [25, 24] is used to emphasize short time horizon performance. Other existing works consider the extreme case when the number of actions is indeed infinite, and tackle such problems with one of two main assumptions: (1) the discovery of a near-optimal/best arm follows some probability measure with *known* parameters [6, 30, 4, 15]; (2) the existence of a *smooth* function represents the mean-payoff over a continuous subset [1, 20, 19, 8, 23, 17]. However, in many situations, neither assumption may be realistic. We make minimal assumptions in this paper. We study the regret minimization problem over a time horizon $T$, which might be unknown, with respect

to a bandit instance with $n$ total arms, out of which $m$ are best/near-optimal arms. We emphasize that the allowed time horizon and the given bandit instance should be viewed as features of *one* problem and together they indicate an intrinsic hardness level. We consider the case when the number of arms $n$ is comparable or larger than the time horizon $T$ so that no standard algorithm provides satisfying result. Our goal is to design algorithms that could adapt to the *unknown* $m$ and achieve optimal regret.

## 1.1 Contributions and paper organization

We make the following contributions. In Section 2, we formally define the regret minimization problem that represents the tension between a very large action space and a limited time horizon; and capture the hardness level in terms of the number of best arms. We provide an adaptive algorithm that is agnostic to the *unknown* number of best arms in Section 3, and theoretically derive its regret bound. In Section 4, we prove a lower bound for our problem setting that indicates that there is no algorithm that can be optimal simultaneously over all hardness levels. Our lower bound also shows that our algorithm provided in Section 3 is Pareto optimal. With additional knowledge of the expected reward of the best arm, in Section 5, we provide an algorithm that achieves the non-adaptive minimax optimal regret, up to polylog factors, without the knowledge of the number of best arms. Experiments conducted in Section 6 confirm our theoretical guarantees and show advantages of our algorithms over previous state-of-the-art. We conclude our paper in Section 7. Most of the proofs are deferred to the Appendix due to lack of space.

## 1.2 Related work

**Time sensitivity and large action space.** As bandit models are getting much more complex, usually with large or infinite action spaces, researchers have begun to pay attention to tradeoffs between regret and time horizons when deploying such models. [13] study a linear bandit problem with ultra-high dimension, and provide algorithms that, under various assumptions, can achieve good reward within short time horizon. [24] also take time horizon into account and model time preference by analyzing a discounted regret. [12] consider a quantile regret minimization problem where they define their regret with respect to expected reward ranked at $(1 - \rho)$-th quantile. One could easily transfer their problem to our setting; however, their regret guarantee is sub-optimal. [18, 4] also consider the problem with $m$ best/near-optimal arms with no other assumptions, but they focus on the pure exploration setting; [4] additionally requires the knowledge of $m$. Another line of research considers the extreme case when the number arms is infinite, but with some *known* regularities. [6] proposes an algorithm with a minimax optimality guarantee under the situation where the reward of each arm follows *strictly* Bernoulli distribution; [27] provides an anytime algorithm that works under the same assumption. [30] relaxes the assumption on Bernoulli reward distribution, however, some other parameters are assumed to be known in their setting.

**Continuum-armed bandit.** Many papers also study bandit problems with continuous action spaces, where they embed each arm $x$ into a bounded subset $\mathcal{X} \subseteq \mathbb{R}^d$ and assume there exists a smooth function $f$ governing the mean-payoff for each arm. This setting is firstly introduced by [1]. When the smoothness parameters are known to the learner or under various assumptions, there exists algorithms [20, 19, 8] with near-optimal regret guarantees. When the smoothness parameters are unknown, however, [23] proves a lower bound indicating no strategy can be optimal simultaneously over all smoothness classes; under extra information, they provide adaptive algorithms with near-optimal regret guarantees. Although achieving optimal regret for all settings is impossible, [17] design adaptive algorithms and prove that they are Pareto optimal. Our algorithms are mainly inspired by the ones in [17, 23]. A closely related line of work [28, 16, 5, 26] aims at minimizing simple regret in the continuum-armed bandit setting.

**Adaptivity to unknown parameters.** [9] argues the awareness of regularity is flawed and one should design algorithms that can *adapt* to the unknown environment. In situations where the goal is pure exploration or simple regret minimization, [18, 28, 16, 5, 26] achieve near-optimal guarantees with unknown regularity because their objectives trade-off exploitation in favor of exploration. In the case of cumulative regret minimization, however, [23] shows no strategy can be optimal simultaneously over all smoothness classes. In special situations or under extra information, [9, 10, 23] provide algorithms that adapt in different ways. [17] borrows the concept of Pareto optimality from economics and provide algorithms with rate functions that are Pareto optimal. Adaptivity is studied in statistics

as well: in some cases, only additional logarithmic factors are required [22, 7]; in others, however, there exists an additional polynomial cost of adaptation [11].

## 2 Problem statement and notation

We consider the multi-armed bandit instance $\underline{\nu} = (\nu_1, \ldots, \nu_n)$ with $n$ probability distributions with means $\mu_i = \mathbb{E}_{X \sim \nu_i}[X] \in [0, 1]$. Let $\mu_\star = \max_{i \in [n]}\{\mu_i\}$ be the highest mean and $S_\star = \{i \in [n] : \mu_i = \mu_\star\}$ denote the subset of best arms.[1] The cardinality $|S_\star| = m$ is *unknown* to the learner. We could also generalize our setting to $S'_\star = \{i \in [n] : \mu_i \geq \mu_\star - \epsilon(T)\}$ with unknown $|S'_\star|$ (i.e., situations where there is an unknown number of near-optimal arms). Setting $\epsilon$ to be dependent on $T$ is to avoid an additive term linear in $T$, e.g., $\epsilon \leq 1/\sqrt{T} \Rightarrow \epsilon T \leq \sqrt{T}$. All theoretical results and algorithms presented in this paper are applicable to this generalized setting with minor modifications. For ease of exposition, we focus on the case with multiple best arms throughout the paper. At each time step $t \in [T]$, the algorithm/learner selects an action $A_t \in [n]$ and receives an independent reward $X_t \sim \nu_{A_t}$. We assume that $X_t - \mu_{A_t}$ is $(1/2)$-sub-Gaussian conditioned on $A_t$.[2] We measure the success of an algorithm through the expected cumulative (pseudo) regret:

$$R_T = T \cdot \mu_\star - \mathbb{E}\left[\sum_{t=1}^{T} \mu_{A_t}\right].$$

We use $\mathcal{R}(T, n, m)$ to denote the set of regret minimization problems with allowed time horizon $T$ and any bandit instance $\underline{\nu}$ with $n$ total arms and $m$ best arms.[3] We emphasize that $T$ is part of the problem instance. We are particularly interested in the case when $n$ is comparable or even larger than $T$, which captures many modern applications where the available action space far exceeds the allowed time horizon. Although learning algorithms may not be able to pull each arm once, one should notice that the true/intrinsic hardness level of the problem could be viewed as $n/m$: selecting a subset uniformly at random with cardinality $\Theta(n/m)$ guarantees, with constant probability, the access to at least one best arm; but of course it is impossible to do this without knowing $m$. We quantify the *intrinsic* hardness level over a set of regret minimization problems $\mathcal{R}(T, n, m)$ as

$$\psi(\mathcal{R}(T, n, m)) = \inf\{\alpha \geq 0 : n/m \leq 2T^\alpha\},$$

where the constant 2 in front of $T^\alpha$ is added to avoid otherwise the trivial case with all best arms when the infimum is 0. $\psi(\mathcal{R}(T, n, m))$ is used here as it captures the *minimax* optimal regret over the set of regret minimization problem $\mathcal{R}(T, n, m)$, as explained later in our review of the MOSS algorithm and the lower bound. As smaller $\psi(\mathcal{R}(T, n, m))$ indicates easier problems, we then define the family of regret minimization problems with hardness level at most $\alpha$ as

$$\mathcal{H}_T(\alpha) = \{\cup \mathcal{R}(T, n, m) : \psi(\mathcal{R}(T, n, m)) \leq \alpha\},$$

with $\alpha \in [0, 1]$. Although $T$ is necessary to define a regret minimization problem, we actually encode the hardness level into a single parameter $\alpha$, which captures the *tension* between the complexity of bandit instance at hand and the allowed time horizon $T$: problems with different time horizons but the same $\alpha$ are equally difficult in terms of the achievable minimax regret (the exponent of $T$). We thus mainly study problems with $T$ large enough so that we could mainly focus on the polynomial terms of $T$. We are interested in designing algorithms with *minimax* guarantees over $\mathcal{H}_T(\alpha)$, but *without* the knowledge of $\alpha$.

MOSS **and upper bound.** In the classical setting, MOSS, designed by [2] and further generalized to the sub-Gaussian case [21] and improved in terms of constant factors [14], achieves the minimax optimal regret. In this paper, we will use MOSS as a subroutine with regret upper bound $O(\sqrt{nT})$ when $T \geq n$. For any problem in $\mathcal{H}_T(\alpha)$ with *known* $\alpha$, one could run MOSS on a subset selected uniformly at random with cardinality $\widetilde{O}(T^\alpha)$ and achieve regret $\widetilde{O}(T^{(1+\alpha)/2})$.

**Lower bound.** The lower bound $\Omega(\sqrt{nT})$ in the classical setting does not work for our setting as its proof heavily relies on the existence of single best arm [21]. However, for problems in $\mathcal{H}_T(\alpha)$, we do have a matching lower bound $\Omega(T^{(1+\alpha)/2})$ as one could always apply the standard lower bound on an bandit instance with $n = \lfloor T^\alpha \rfloor$ and $m = 1$. For general value of $m$, a lower bound of the order $\Omega(\sqrt{T(n-m)/m}) = \Omega(T^{(1+\alpha)/2})$ for the $m$-best arms case could be obtained following similar analysis in Chapter 15 of [21].

Although $\log T$ may appear in our bounds, throughout the paper, we focus on problems with $T \geq 2$ as otherwise the bound is trivial.

## 3 An adaptive algorithm

Algorithm 1 takes time horizon $T$ and a user-specified $\beta \in [1/2, 1]$ as input, and it is mainly inspired by [17]. Algorithm 1 operates in iterations with geometrically-increasing length $\Delta T_i = 2^{p+i}$ with $p = \lceil \log_2 T^\beta \rceil$. At each iteration $i$, it restarts MOSS on a set $S_i$ consisting of $K_i = 2^{p+2-i}$ real arms selected uniformly at random *plus* a set of "virtual" *mixture-arms* (one from each of the $1 \leq j < i$ previous iterations, none if $i = 1$). The mixture-arms are constructed as follows. After each iteration $i$, let $\widehat{p}_i$ denote the vector of empirical sampling frequencies of the arms in that iteration (i.e., the $k$-th element of $\widehat{p}_i$ is the number of times arm $k$, including all previously constructed mixture-arms, was sampled in iteration $i$ divided by the total number of samples $\Delta T_i$). The mixture-arm for iteration $i$ is the $\widehat{p}_i$-mixture of the arms, denoted by $\widetilde{\nu}_i$. When MOSS samples from $\widetilde{\nu}_i$ it first draws $i_t \sim \widehat{p}_i$, then draws a sample from the corresponding arm $\nu_{i_t}$ (or $\widetilde{\nu}_{i_t}$). The mixture-arms provide a convenient summary of the information gained in the previous iterations, which is key to our theoretical analysis. Although our algorithm is working on fewer regular arms in later iterations, information summarized in mixture-arms is good enough to provide guarantees. We name our algorithm MOSS++ as it restarts MOSS at each iteration with past information summarized in mixture-arms. We provide an anytime version of Algorithm 1 in Appendix A.2 via the standard doubling trick.

---

**Algorithm 1:** MOSS++

**Input:** Time horizon $T$ and user-specified parameter $\beta \in [1/2, 1]$.
1: **Set:** $p = \lceil \log_2 T^\beta \rceil$, $K_i = 2^{p+2-i}$ and $\Delta T_i = \min\{2^{p+i}, T\}$.
2: **for** $i = 1, \ldots, p$ **do**
3:     Run MOSS on a subset of arms $S_i$ for $\Delta T_i$ rounds. $S_i$ contains $K_i$ real arms selected uniformly at random *and* the set of virtual mixture-arms from previous iterations, i.e., $\{\widetilde{\nu}_j\}_{j<i}$.
4:     Construct a virtual mixture-arm $\widetilde{\nu}_i$ based on empirical sampling frequencies of MOSS above.
5: **end for**

---

### 3.1 Analysis and discussion

We use $\mu_S = \max_{\nu \in S}\{\mathbb{E}_{X \sim \nu}[X]\}$ to denote the highest expected reward over a set of distributions/arms $S$. For any algorithm that only works on $S$, we can decompose the regret into approximation error and learning error:

$$R_T = \underbrace{T \cdot (\mu_\star - \mu_S)}_{\text{approximation error due to the selection of } S} + \underbrace{T \cdot \mu_S - \mathbb{E}\left[\sum_{t=1}^T \mu_{A_t}\right]}_{\text{learning error due to the sampling rule } \{A_t\}_{t=1}^T} . \quad (1)$$

This type of regret decomposition was previously used in [20, 3, 17] to deal with the continuum-armed bandit problem. We consider here a probabilistic version, with randomness in the selection of $S$, for the classical setting.

The main idea behind providing guarantees for MOSS++ is to decompose its regret at each iteration, using Eq. (1), and then bound the expected approximation error and learning error separately. The expected learning error at each iteration could always be controlled as $\widetilde{O}(T^\beta)$ thanks to regret guarantees for MOSS and specifically chosen parameters $p$, $K_i$, $\Delta T_i$. Let $i_\star$ be the largest integer such that $K_i \geq 2T^\alpha \log \sqrt{T}$ still holds. The expected approximation error in iteration $i \leq i_\star$ could be

upper bounded by $\sqrt{T}$ following an analysis on hypergeometric distribution. As a result, the expected regret in iteration $i \leq i_\star$ is $\widetilde{O}(T^\beta)$. Since the mixture-arm $\widetilde{\nu}_{i_\star}$ is included in all following iterations, we could further bound the expected approximation error in iteration $i > i_\star$ by $\widetilde{O}(T^{1+\alpha-\beta})$ after a careful analysis on $\Delta T_i / \Delta T_{i_\star}$. This intuition is formally stated and proved in Theorem 1.

**Theorem 1.** *Run* MOSS++ *with time horizon $T$ and an user-specified parameter $\beta \in [1/2, 1]$ leads to the following regret upper bound:*

$$\sup_{\omega \in \mathcal{H}_T(\alpha)} R_T \leq C \left(\log_2 T\right)^{5/2} \cdot T^{\min\{\max\{\beta, 1+\alpha-\beta\}, 1\}},$$

*where $C$ is a universal constant.*

**Remark 1.** *We primarily focus on the polynomial terms in $T$ when deriving the bound, but put no effort in optimizing the polylog term. The $5/2$ exponent of $\log_2 T$ might be tightened as well.*

The theoretical guarantee is closely related to the user-specified parameter $\beta$: when $\beta > \alpha$, we suffer a multiplicative cost of adaptation $\widetilde{O}(T^{|(2\beta-\alpha-1)/2|})$, with $\beta = (1+\alpha)/2$ hitting the sweet spot, comparing to non-adaptive minimax regret; when $\beta \leq \alpha$, there is essentially no guarantees. One may hope to improve this result. However, our analysis in Section 4 indicates: (1) achieving minimax optimal regret for all settings simultaneously is *impossible*; and (2) the rate function achieved by MOSS++ is already *Pareto optimal*.

# 4   Lower bound and Pareto optimality

## 4.1   Lower bound

In this section, we show that designing algorithms with the non-adaptive minimax optimal guarantee over all values of $\alpha$ is impossible. We first state the result in the following general theorem.

**Theorem 2.** *For any $0 \leq \alpha' < \alpha \leq 1$, assume $T^\alpha \leq B$ and $\lfloor T^\alpha \rfloor - 1 \geq \max\{T^\alpha/4, 2\}$. If an algorithm is such that $\sup_{\omega \in \mathcal{H}_T(\alpha')} R_T \leq B$, then the regret of this algorithm is lower bounded on $\mathcal{H}_T(\alpha)$:*

$$\sup_{\omega \in \mathcal{H}_T(\alpha)} R_T \geq 2^{-10} T^{1+\alpha} B^{-1}. \tag{2}$$

To give an interpretation of Theorem 2, we consider any algorithm/policy $\pi$ together with regret minimization problems $\mathcal{H}_T(\alpha')$ and $\mathcal{H}_T(\alpha)$ satisfying corresponding requirements. On one hand, if algorithm $\pi$ achieves a regret that is order-wise larger than $\widetilde{O}(T^{(1+\alpha')/2})$ over $\mathcal{H}_T(\alpha')$, it is already not minimax optimal for $\mathcal{H}_T(\alpha')$. Now suppose $\pi$ achieves a near-optimal regret, i.e., $\widetilde{O}(T^{(1+\alpha')/2})$, over $\mathcal{H}_T(\alpha')$; then, according to Eq. (2), $\pi$ must incur a regret of order at least $\widetilde{\Omega}(T^{1/2+\alpha-\alpha'/2})$ on one problem in $\mathcal{H}_T(\alpha')$. This, on the other hand, makes algorithm $\pi$ strictly sub-optimal over $\mathcal{H}_T(\alpha)$.

## 4.2   Pareto optimality

We capture the performance of any algorithm by its dependence on polynomial terms of $T$ in the asymptotic sense. Note that the hardness level of a problem is encoded in $\alpha$.

**Definition 1.** *Let $\theta : [0, 1] \to [0, 1]$ denote a non-decreasing function. An algorithm achieves the rate function $\theta$ if*

$$\forall \epsilon > 0, \forall \alpha \in [0, 1], \quad \limsup_{T \to \infty} \frac{\sup_{\omega \in \mathcal{H}_T(\alpha)} R_T}{T^{\theta(\alpha)+\epsilon}} < +\infty.$$

Recall that a function $\theta'$ is strictly smaller than another function $\theta$ in pointwise order if $\theta'(\alpha) \leq \theta(\alpha)$ for all $\alpha$ and $\theta'(\alpha_0) < \theta(\alpha_0)$ for at least one value of $\alpha_0$. As there may not always exist a pointwise ordering over rate functions, following [17], we consider the notion of Pareto optimality over rate functions achieved by some algorithms.

**Definition 2.** *A rate function $\theta$ is Pareto optimal if it is achieved by an algorithm, and there is no other algorithm achieving a strictly smaller rate function $\theta'$ in pointwise order. An algorithm is Pareto optimal if it achieves a Pareto optimal rate function.*

Combining the results in Theorem 1 and Theorem 2 with above definitions, we could further obtain the following result in Theorem 3.

**Theorem 3.** *The rate function achieved by* MOSS++ *with any $\beta \in [1/2, 1]$, i.e.,*

$$\theta_\beta : \alpha \mapsto \min\{\max\{\beta, 1 + \alpha - \beta\}, 1\}, \tag{3}$$

*is Pareto optimal.*

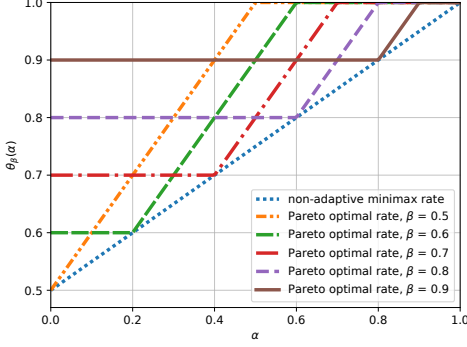

Figure 1: Pareto optimal rates

Fig. 1 provides an illustration of the rate functions achieved by MOSS++ with different $\beta$ as input, as well as the non-adaptive minimax optimal rate.

**Remark 2.** *One should notice that the naive algorithm running* MOSS *on a subset selected uniformly at random with cardinality $\widetilde{O}(T^{\beta'})$ is not Pareto optimal, since running* MOSS++ *with $\beta = (1 + \beta')/2$ leads to a strictly smaller rate function. The algorithm provided in [12], if transferred to our setting and allowing time horizon dependent quantile, is not Pareto optimal as well since it corresponds to the rate function $\theta(\alpha) = \max\{2.89\,\alpha, 0.674\}$.*

## 5 Learning with extra information

Although previous Section 4 gives negative results on designing algorithms that could optimally adapt to all settings, one could actually design such an algorithm *with* extra information. In this section, we provide an algorithm that takes the expected reward of the best arm $\mu_\star$ (or an estimated one with error up to $1/\sqrt{T}$) as extra information, and achieves near minimax optimal regret over all settings simultaneously. Our algorithm is mainly inspired by [23].

### 5.1 Algorithm

We name our Algorithm 3 `Parallel` as it maintains $\lceil \log T \rceil$ instances of subroutine, i.e., Algorithm 2, in parallel. Each subroutine $\mathtt{SR}_i$ is initialized with time horizon $T$ and hardness level $\alpha_i = i/\lceil \log T \rceil$. We use $T_{i,t}$ to denote the number of samples allocated to $\mathtt{SR}_i$ up to time $t$, and represent its empirical regret at time $t$ as $\widehat{R}_{i,t} = T_{i,t} \cdot \mu_\star - \sum_{t=1}^{T_{i,t}} X_{i,t}$ with $X_{i,t} \sim \nu_{A_{i,t}}$ being the $t$-th empirical reward obtained by $\mathtt{SR}_i$ and $A_{i,t}$ being the index of the $t$-th arm pulled by $\mathtt{SR}_i$.

---

**Algorithm 2:** MOSS Subroutine (SR)

---

**Input:** Time horizon $T$ and hardness level $\alpha$.
1: Select a subset of arms $S_\alpha$ uniformly at random with $|S_\alpha| = \lceil 2T^\alpha \log \sqrt{T} \rceil$ and run MOSS on $S_\alpha$.

---

`Parallel` operates in iterations of length $\lceil \sqrt{T} \rceil$. At the beginning of each iteration, i.e., at time $t = i \cdot \lceil \sqrt{T} \rceil$ for $i \in \{0\} \cup [[\sqrt{T}] - 1]$, `Parallel` first selects the subroutine with the lowest (breaking ties arbitrarily) empirical regret so far, i.e., $k = \arg\min_{i \in [[\log T]]} \widehat{R}_{i,t}$; it then *resumes* the learning process of $\mathtt{SR}_k$, from where it halted, for another $\lceil \sqrt{T} \rceil$ more pulls. All the information is updated at the end of that iteration. An anytime version of Algorithm 3 is provided in Appendix C.3.

### 5.2 Analysis

As `Parallel` discretizes the hardness parameter over a grid with interval $1/\lceil \log T \rceil$, we first show that running the best subroutine alone leads to regret $\widetilde{O}(T^{(1+\alpha)/2})$.

**Algorithm 3:** `Parallel`

---

**Input:** Time horizon $T$ and the optimal reward $\mu_\star$.

1: **set:** $p = \lceil \log T \rceil$, $\Delta = \lceil \sqrt{T} \rceil$ and $t = 0$.
2: **for** $i = 1, \ldots, p$ **do**
3:     Set $\alpha_i = i/p$, initialize $\mathrm{SR}_i$ with $\alpha_i$, $T$; set $T_{i,t} = 0$, and $\widehat{R}_{i,t} = 0$.
4: **end for**
5: **for** $i = 1, \ldots, \Delta - 1$ **do**
6:     Select $k = \arg\min_{i \in [p]} \widehat{R}_{i,t}$ and run $\mathrm{SR}_k$ for $\Delta$ rounds.
7:     Update $T_{k,t} = T_{k,t} + \Delta$, $\widehat{R}_{k,t} = T_{k,t} \cdot \mu_\star - \sum_{t=1}^{T_{k,t}} X_{k,t}$, $t = t + \Delta$.
8: **end for**

---

**Lemma 1.** *Suppose $\alpha$ is the true hardness parameter and $\alpha_i - 1/\lceil \log T \rceil < \alpha \leq \alpha_i$, run Algorithm 2 with time horizon $T$ and $\alpha_i$ leads to the following regret bound:*

$$\sup_{\omega \in \mathcal{H}_T(\alpha)} R_T \leq C \, \log T \cdot T^{(1+\alpha)/2},$$

*where $C$ is a universal constant.*

Since `Parallel` always allocates new samples to the subroutine with the lowest empirical regret so far, we know that the regret of every subroutine should be roughly of the same order at time $T$. In particular, all subroutines should achieve regret $\widetilde{O}(T^{(1+\alpha)/2})$, as the best subroutine does. `Parallel` then achieves the non-adaptive minimax optimal regret, up to polylog factors, *without* knowing the true hardness level $\alpha$.

**Theorem 4.** *For any $\alpha \in [0, 1]$ unknown to the learner, run `Parallel` with time horizon $T$ and optimal expected reward $\mu_\star$ leads to the following regret upper bound:*

$$\sup_{\omega \in \mathcal{H}_T(\alpha)} R_T \leq C \, (\log T)^2 \, T^{(1+\alpha)/2},$$

*where $C$ is a universal constant.*

## 6 Experiments

We conduct three experiments to compare our algorithms with baselines. In Section 6.1, we compare the performance of each algorithm on problems with varying hardness levels. We examine how the regret curve of each algorithm increases on synthetic and real-world datasets in Section 6.2 and Section 6.3, respectively.

We first introduce the nomenclature of the algorithms. We use `MOSS` to denote the standard `MOSS` algorithm; and `MOSS Oracle` to denote Algorithm 2 with *known* $\alpha$. `Quantile` represents the algorithm (QRM2) proposed by [12] to minimize the regret with respect to the $(1 - \rho)$-th quantile of means among arms, without the knowledge of $\rho$. One could easily transfer `Quantile` to our settings with top-$\rho$ fraction of arms treated as best arms. As suggested in [12], we reuse the statistics obtained in previous iterations of `Quantile` to improve its sample efficiency. We use `MOSS++` to represent the vanilla version of Algorithm 1; and use `empMOSS++` to represent an empirical version such that: (1) `empMOSS++` reuse statistics obtained in previous round, as did in `Quantile`; and (2) instead of selecting $K_i$ real arms uniformly at random at the $i$-th iteration, `empMOSS++` selects $K_i$ arms with the highest empirical mean for $i > 1$. We choose $\beta = 0.5$ for `MOSS++` and `empMOSS++` in all experiments.[4] All results are averaged over 100 experiments. Shaded area represents 0.5 standard deviation for each algorithm.

### 6.1 Adaptivity to hardness level

We compare our algorithms with baselines on regret minimization problems with different hardness levels. For this experiment, we generate best arms with expected reward 0.9 and sub-optimal arms

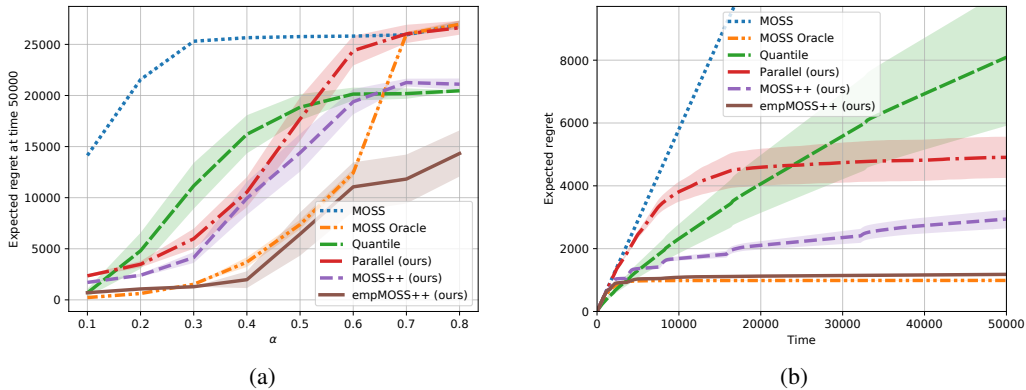

Figure 2: (a) Comparison with varying hardness levels (b) Regret curve comparison with $\alpha = 0.25$.

with expected reward evenly distributed among $\{0.1, 0.2, 0.3, 0.4, 0.5\}$. All arms follow Bernoulli distribution. We set the time horizon to $T = 50000$ and consider the total number of arms $n = 20000$. We vary $\alpha$ from 0.1 to 0.8 (with interval 0.1) to control the number of best arms $m = \lceil n/2T^\alpha \rceil$ and thus the hardness level. In Fig. 2(a), the regret of any algorithm gets larger as $\alpha$ increases, which is expected. MOSS does not provide satisfying performance due to the large action space and the relatively small time horizon. Although implemented in an anytime fashion, Quantile could be roughly viewed as an algorithm that runs MOSS on a subset selected uniformly at random with cardinality $T^{0.347}$. Quantile displays good performance when $\alpha = 0.1$, but suffers regret much worse than MOSS++ and empMOSS++ when $\alpha$ gets larger. Note that the regret curve of Quantile gets flattened at 20000 is expected: it simply learns the best sub-optimal arm and suffers a regret $50000 \times (0.9 - 0.5)$. Although Parallel enjoys near minimax optimal regret, the regret it suffers from is the summation of 11 subroutines, which hurts its empirical performance. empMOSS++ achieves performance comparable to MOSS Oracle when $\alpha$ is small, and achieve the best empirical performance when $\alpha \geq 0.3$. When $\alpha \geq 0.7$, MOSS Oracle needs to explore most/all of the arms to statistically guarantee the finding of at least one best arm, which hurts its empirical performance.

## 6.2 Regret curve comparison

We compare how the regret curve of each algorithm increases in Fig. 2(b). We consider the same regret minimization configurations as described in Section 6.1 with $\alpha = 0.25$. empMOSS++, MOSS++ and Parallel all outperform Quantile with empMOSS++ achieving the performance closest to MOSS Oracle. MOSS Oracle, Parallel and empMOSS++ have flattened their regret curve indicating they could confidently recommend the best arm. The regret curves of MOSS++ and Quantile do not flat as the random-sampling component in each of their iterations encourage them to explore new arms. Comparing to MOSS++, Quantile keeps increasing its regret at a much faster rate and with a much larger variance, which empirically confirms the sub-optimality of their regret guarantees.

## 6.3 Real-world dataset

We also compare all algorithms in a realistic setting of recommending funny captions to website visitors. We use a real-world dataset from the *New Yorker Magazine* Cartoon Caption Contest[5]. The dataset of 1-3 star caption ratings/rewards for Contest 652 consists of $n = 10025$ captions[6]. We use the ratings to compute Bernoulli reward distributions for each caption as follows. The mean of each caption/arm $i$ is calculated as the percentage $p_i$ of its ratings that were funny or somewhat funny (i.e., 2 or 3 stars). We normalize each $p_i$ with the best one and then threshold each: if $p_i \geq 0.8$, then put $p_i = 1$; otherwise leave $p_i$ unaltered. This produces a set of $m = 54$ best arms with rewards 1 and all

other 9971 arms with rewards among $[0, 0.8]$. We set $T = 10^5$ and this results in a hardness level around $\alpha \approx 0.43$.

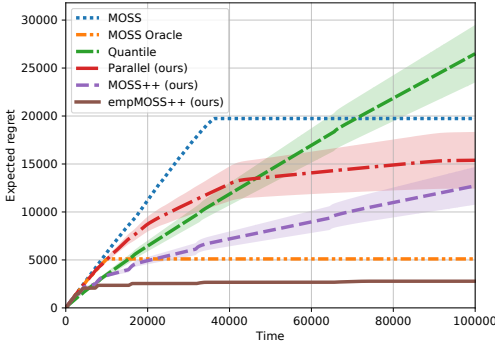

Figure 3: Regret comparison with real-world dataset

Using these Bernoulli reward models, we compare the performance of each algorithm, as shown in Fig. 3. `MOSS`, `MOSS Oracle`, `Parallel` and `empMOSS++` have flattened their regret curve indicating they could confidently recommend the funny captions (i.e., best arms). Although `MOSS` could eventually identify a best arm in this problem, its cumulative regret is more than 7x of the regret achieved by `empMOSS++` due to its initial exploration phase. The performance of `Quantile` is even worse, and its cumulative regret is more than 9x of the regret achieved by `empMOSS++`. One surprising phenomenon is that `empMOSS++` outperforms `MOSS Oracle` in this realistic setting. Our hypothesis is that `MOSS Oracle` is a little bit conservative and selects an initial set with cardinality too large. This experiment demonstrates the effectiveness of `empMOSS++` and `MOSS++` in modern applications of bandit algorithm with large action space and limited time horizon.

## 7 Conclusion

We study a regret minimization problem with large action space but limited time horizon, which captures many modern applications of bandit algorithms. Depending on the number of best/near-optimal arms, we encode the hardness level, in terms of minimax regret achievable, of the given regret minimization problem into a single parameter $\alpha$, and we design algorithms that could adapt to this *unknown* hardness level. Our first algorithm `MOSS++` takes a user-specified parameter $\beta$ as input and provides guarantees as long as $\alpha < \beta$; our lower bound further indicates the rate function achieved by `MOSS++` is Pareto optimal. Although no algorithm can achieve near minimax optimal regret over all $\alpha$ simultaneously, as demonstrated by our lower bound, we overcome this limitation with an (often) easily-obtained extra information and propose `Parallel` that is near-optimal for all settings. Inspired by `MOSS++`, We also propose `empMOSS++` with excellent empirical performance. Experiments on both synthetic and real-world datasets demonstrate the efficiency of our algorithms over the previous state-of-the-art.

## Broader Impact

This paper provides efficient algorithms that work well in modern applications of bandit algorithms with large action space but limited time horizon. We make *minimal* assumption about the setting, and our algorithms can automatically adapt to *unknown* hardness levels. Worst-case regret guarantees are provided for our algorithms; we also show `MOSS++` is Pareto optimal and `Parallel` is minimax optimal, up to polylog factors. `empMOSS++` is provided as a practical version of `MOSS++` with excellent empirical performance. Our algorithms are particularly useful in areas such as e-commence and movie/content recommendation, where the action space is enormous but possibly contains multiple best/satisfactory actions. If deployed, our algorithms could automatically adapt to the hardness level of the recommendation task and benefit both service-providers and customers through efficiently delivering satisfactory content. One possible negative outcome is that items recommended to a *specific* user/customer might only come from a subset of the action space. However, this is *unavoidable* when the number of items/actions exceeds the allowed time horizon. In fact, one should notice that all items/actions will be selected with essentially the same probability, thanks to the incorporation of random selection processes in our algorithms. Our algorithms will *not* leverage/create biases due to the same reason. Overall, we believe this paper's contribution will have a net positive impact.

## Acknowledgments and Disclosure of Funding

The authors would like to thank anonymous reviewers for their comments and suggestions. This work was partially supported by NSF grant no. 1934612.

## Footnotes

[1] Throughout the paper, we denote by $[K]$ the set $\{1, \ldots, K\}$ for any positive integer $K$.

[2] We say a random variable $X$ is $\sigma$-sub-Gaussian if $\mathbb{E}[\exp(\lambda X)] \leq \exp(\sigma^2 \lambda^2/2)$ for all $\lambda \in \mathbb{R}$.

[3] Our setting could be generalized to the case with infinite arms: one can consider embedding arms into an arm space $\mathcal{X}$ and let $p$ be the probability that an arm sampled uniformly at random is (near-)optimal. $1/p$ will then serve a similar role as $n/m$ does in the original definition.

[4]Increasing $\beta$ generally leads to worse performance on problems with small $\alpha$ but better performance on problems with large $\alpha$.

[5]https://www.newyorker.com/cartoons/contest.

[6]Available online at https://nextml.github.io/caption-contest-data.

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
