[Supplementary Material]

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

[7]One can remove the $(\log_2 T)^{1/2}$ term in many cases, e.g., when $\beta > 1/2$ and $T$ is large enough (with respect to $\beta$). However, we mainly focus on the polynomial terms here.

[8] $K \geq 2$ holds for $T$ large enough.

[9]One can sharpen the $\log T$ term to $(\log T)^{1/2}$ in many cases, e.g., when $\alpha < 1$ and $T$ is large enough (with respect to $\alpha$). Again, we mainly focus on the polynomial terms here.

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

# A    Omitted proofs for Section 3

We introduce the notation $R_{T|\mathcal{F}} = T \cdot \mu_\star - \mathbb{E}[\sum_{t=1}^T X_t | \mathcal{F}]$ for any $\sigma$-algebra. One should also notice that $\mathbb{E}[R_{T|\mathcal{F}}] = R_T$.

## A.1    Proof of Theorem 1

**Lemma 2.** *For an instance with $n$ total arms and $m$ best arms, and for a subset $S$ selected uniformly at random with cardinality $k$, the probability that none of the best arms are selected in $S$ is upper bounded by $\exp(-mk/n)$.*

*Proof.* Consider selecting $k$ items out of $n$ items without replacement; and suppose there are $m$ target items. Let $\mathcal{E}$ denote the event where none of the target items are selected, we then have

$$
\begin{aligned}
\mathbb{P}\left(\mathcal{E}\right) = \frac{\binom{n-m}{k}}{\binom{n}{k}} &= \frac{\frac{(n-m)!}{(n-m-k)!k!}}{\frac{n!}{(n-k)!k!}} \\
&= \frac{(n-m)!}{(n-m-k)!} \cdot \frac{(n-k)!}{n!} \\
&= \prod_{i=0}^{k-1} \frac{n-m-i}{n-i} \\
&\leq \left(\frac{n-m}{n}\right)^k \qquad\qquad (4) \\
&\leq \exp\left(-\frac{m}{n} \cdot k\right), \qquad\qquad (5)
\end{aligned}
$$

where Eq. (4) comes from the fact that $\frac{n-m-i}{n-i}$ is decreasing in $i$; and Eq. (5) comes from the fact that $1 - x \leq \exp(-x)$ for all $x \in \mathbb{R}$.

Selecting arms with replacement gives the same guarantee (which directly goes to Eq. (4)), and can be used in corner cases when $k > n$. $\qquad\square$

**Theorem 1.** *Run* MOSS++ *with time horizon $T$ and an user-specified parameter $\beta \in [1/2, 1]$ leads to the following regret upper bound:*

$$
\sup_{\omega \in \mathcal{H}_T(\alpha)} R_T \leq C \left(\log_2 T\right)^{5/2} \cdot T^{\min\{\max\{\beta, 1+\alpha-\beta\}, 1\}},
$$

*where $C$ is a universal constant.*

*Proof.* Let $T_i = \sum_{j=1}^i \Delta T_j$. We first notice that Algorithm 1 is a valid algorithm in the sense that it selects an arm $A_t$ for any $t \in [T]$, i.e., it does not terminate before time $T$: the argument is clearly true if there exists $i \in [p]$ such that $\Delta T_i = T$; otherwise, we can show that

$$
T_p = \sum_{i=1}^p \Delta T_i = 2(2^{2p} - 1) \geq 2^{2p} \geq T,
$$

for all $\beta \in [1/2, 1]$.

We will only consider the case when $\alpha < \beta$ in the following since otherwise Theorem 1 trivially holds due to $T^{1+\alpha-\beta} \geq T$.

Let $\mathcal{F}_{i-1}$ represents the information available at the beginning of iteration $i$, including the random selection process of generating $S_i$. We denote $R_{\Delta T_i} = \Delta T_i \cdot \mu_\star - \mathbb{E}[\sum_{t=T_{i-1}+1}^{T_i} X_t]$ the expected cumulative regret at iteration $i$. Recall that we use $R_{\Delta T_i | \mathcal{F}_{i-1}}$ to represent the expected regret conditioned on $\mathcal{F}_{i-1}$ and have $\mathbb{E}[R_{\Delta T_i | \mathcal{F}_{i-1}}] = R_{\Delta T_i}$; we also use $\mu_{S_i} = \max_{\nu \in S_i}\{\mathbb{E}_{X \sim \nu}[X]\}$. Applying Eq. (1) on $R_{\Delta T_i | \mathcal{F}_{i-1}}$ leads to

$$
R_{\Delta T_i | \mathcal{F}_{i-1}} = \Delta T_i \cdot (\mu_\star - \mu_{S_i}) + \left(\Delta T_i \cdot \mu_{S_i} - \mathbb{E}\left[\sum_{t=T_{i-1}+1}^{T_i} \mu_{A_t} \,\middle|\, \mathcal{F}_{i-1}\right]\right), \qquad (6)
$$

where, by a slightly abuse of notations, we use $\mu_{A_t}$ to refer to the mean of arm $A_t \in S_i$, which could also be the mean of a virtual arm constructed in one of the previous iterations.

Note that we will be working on a bandit instance with $(\sqrt{2}/2)$-sub-Gaussian noise when the extra virtual arms are included: let $X$ be a sample from a virtual mixture-arm $\widetilde{\nu}_i$, which is realized by first sampling an index $i_t$ (of a real arm) from the empirical measure, and then draw $X$ from the real arm $\nu_{i_t}$. We have $X - \mathbb{E}[X] = (X - \mu_{i_t}) + (\mu_{i_t} - \mathbb{E}[X])$ and thus for any $\lambda \in \mathbb{R}$,

$$
\begin{aligned}
\mathbb{E}\left[\exp\left(\lambda\left(X - \mathbb{E}[X]\right)\right)\right] &= \mathbb{E}\left[\mathbb{E}\left[\exp\left(\lambda\left(X - \mathbb{E}[X]\right)\right)\middle|i_t\right]\right] \\
&= \mathbb{E}\left[\exp\left(\lambda(\mu_{i_t} - \mathbb{E}[X])\right)\mathbb{E}\left[\exp\left(\lambda\left(X - \mu_{i_t}\right)\right)\middle|i_t\right]\right] \\
&\leq \exp\left(\frac{\lambda^2/4}{2}\right)\mathbb{E}\left[\exp\left(\lambda(\mu_{i_t} - \mathbb{E}[X])\right)\right] \\
&\leq \exp\left(\frac{\lambda^2/4}{2} + \frac{\lambda^2/4}{2}\right) \quad\quad\quad (7)\\
&= \exp\left(\frac{\lambda^2/2}{2}\right)
\end{aligned}
$$

where Eq. (7) comes from the fact that $\mu_{i_t} \in [0,1]$ and $\mathbb{E}[\mu_{i_t}] = \mathbb{E}[X]$. In the following, we'll directly plug in the regret bound of MOSS for the 1-sub-Gaussian case.

We first consider the learning error for any iteration $i \in [p]$. Although $\mu_{S_i}$ is random, it is fixed at time $T_{i-1} + 1$ [17]. Since MOSS restarts at each iteration, conditioning on the information available at the beginning of the $i$-th iteration, i.e., $\mathcal{F}_{i-1}$, and apply the regret bound for MOSS , we have:

$$
\Delta T_i \cdot \mu_{S_i} - \mathbb{E}\left[\sum_{t=T_{i-1}+1}^{T_i} \mu_{A_t} \middle| \mathcal{F}_{i-1}\right] \leq 39\sqrt{|S_i|\Delta T_i} + |S_i| \quad\quad\quad (8)
$$

$$
\begin{aligned}
&= 39\sqrt{(K_i + i - 1)\Delta T_i} + (K_i + i - 1) \\
&\leq 39\sqrt{K_i\Delta T_i + (p-1)\Delta T_i} + (K_i + p - 1) \quad\quad\quad (9)\\
&\leq 39\sqrt{2^{2p+2} + (p-1)T} + 2^{p+1} + (p-1) \quad\quad\quad (10)\\
&\leq 39\sqrt{16T^{2\beta} + \log_2(T^\beta)\,T} + 4T^\beta + \log_2 T^\beta \quad\quad\quad (11)\\
&\leq 166\left(\log_2 T\right)^{1/2} \cdot T^\beta, \quad\quad\quad (12)
\end{aligned}
$$

where Eq. (8) comes from the guarantee of MOSS [21]; Eq. (9) comes from $i \leq p$; Eq. (10) comes from the definition of $K_i$ and $\Delta T_i$; Eq. (11) comes from the fact that $p = \lceil \log_2 T^\beta \rceil \leq \log_2 T^\beta + 1$; Eq. (12) comes from some trivial boundings on the constant.[7]

Taking expectation over all randomness on Eq. (6), we obtain

$$
R_{\Delta T_i} \leq \Delta T_i \cdot \mathbb{E}\left[(\mu_\star - \mu_{S_i})\right] + 166\left(\log_2 T\right)^{1/2} \cdot T^\beta. \quad\quad\quad (13)
$$

Now, we only need to consider the first term, i.e., the expected approximation error over the $i$-th iteration. Let $\mathcal{E}_i$ denote the event that none of the best arms, among regular arms, is selected in $S_i$, according to Lemma 2, we further have

$$
\begin{aligned}
\Delta T_i \cdot \mathbb{E}\left[(\mu_\star - \mu_{S_i})\right] &\leq \Delta T_i \cdot (0 \cdot \mathbb{P}(\neg\mathcal{E}_i) + 1 \cdot \mathbb{P}(\mathcal{E}_i)) \quad\quad\quad (14)\\
&\leq \Delta T_i \cdot \exp(-K_i/(2T^\alpha)), \quad\quad\quad (15)
\end{aligned}
$$

where we use the fact the $\mu_i \in [0,1]$ in Eq. (14); and directly plug $n/m \leq 2T^\alpha$ into Eq. (5) to get Eq. (15).

Let $i_\star \in [p]$ be the largest integer, if exists, such that $K_{i_\star} \geq 2T^\alpha \log\sqrt{T}$, we then have that, for any $i \leq i_\star$,

$$
\Delta T_i \cdot \mathbb{E}\left[(\mu_\star - \mu_{S_i})\right] \leq \Delta T_i/\sqrt{T} \leq T/\sqrt{T} \leq \sqrt{T}. \quad\quad\quad (16)
$$

Note that this choice of $i_\star$ indicates $T^\alpha \log T \le K_{i_\star} < 2T^\alpha \log T$.

If we have $K_1 < 2T^\alpha \log \sqrt{T}$, we then set $i_\star = 1$. Notice that $K_1 = 2^{p+1} = 2^{\lceil \log_2 T^\beta \rceil + 1} \ge 2T^\beta > 2T^\alpha$, we then have

$$\Delta T_1 \cdot \mathbb{E}\left[(\mu_\star - \mu_{S_1})\right] \le \Delta T_1 \exp(-1) \le 2^{p+1} \exp(-1) < 2T^\beta. \tag{17}$$

Combining Eq. (13) with Eq. (16) or Eq. (17), we have for any $i \le i_\star$, and in particular for $i = i_\star$,

$$R_{\Delta T_i} \le \max\{\sqrt{T}, 2T^\beta\} + 166 \, (\log_2 T)^{1/2} \cdot T^\beta$$
$$\le 168 \, (\log_2 T)^{1/2} \cdot T^\beta. \tag{18}$$

In the case when $i_\star = p$ or when $\Delta T_{i_\star} = \min\{2^{p+i}, T\} = T$, we know that MOSS++ will in fact stop at a time step no larger than $T_{i_\star}$ (since the allowed time horizon is $T$), and incur no regret in iterations $i > i_\star$. In the following, we only consider the case when $i_\star < p$ and $\Delta T_{i_\star} = 2^{p+i_\star}$. As a result, we have $K_{i_\star} \Delta T_{i_\star} = 2^{2p+2}$ and thus

$$\Delta T_{i_\star} = \frac{2^{2p+2}}{K_{i_\star}} > \frac{2^{2p+1}}{T^\alpha \log T}, \tag{19}$$

where Eq. (19) comes from the fact that $K_{i_\star} < \max\{2T^\alpha \log T, 2T^\alpha \log \sqrt{T}\} = 2T^\alpha \log T$ by definition of $i_\star$.

We now analysis the expected approximation error for iteration $i > i_\star$. Since the sampling information during the $i_\star$-th iteration is summarized in the virtual mixture-arm $\widetilde{\nu}_{i_\star}$, and being added to all $S_i$ for all $i > i_\star$. Let $\widetilde{\mu}_{i_\star} = \mathbb{E}_{X \sim \widetilde{\nu}_{i_\star}}[X]$ denote the expected reward of sampling according to the virtual mixture-arm $\widetilde{\nu}_{i_\star}$. For any $i > i_\star$, we then have

$$\Delta T_i \cdot \mathbb{E}\left[(\mu_\star - \mu_{S_i})\right] \le \Delta T_i \cdot (\mu_\star - \mathbb{E}[\widetilde{\mu}_{i_\star}])$$
$$= \frac{\Delta T_i}{\Delta T_{i_\star}} \cdot (\Delta T_{i_\star} \cdot (\mu_\star - \mathbb{E}[\widetilde{\mu}_{i_\star}]))$$
$$= \frac{\Delta T_i}{\Delta T_{i_\star}} \cdot \left(\Delta T_{i_\star} \cdot \mu_\star - \sum_{t=T_{i_\star-1}+1}^{T_{i_\star}} \mathbb{E}[\mu_{A_t}]\right)$$
$$= \frac{\Delta T_i}{\Delta T_{i_\star}} \cdot R_{\Delta T_{i_\star}}$$
$$< \frac{\Delta T_i}{\frac{2^{2p+1}}{T^\alpha \log T}} \cdot 168 \, (\log_2 T)^{1/2} \cdot T^\beta$$
$$\le \frac{T^{1+\alpha+\beta}}{2^{2p}} \cdot 84 \, (\log_2 T)^{3/2} \tag{20}$$
$$\le 84 \, (\log_2 T)^{3/2} \cdot T^{1+\alpha-\beta}, \tag{21}$$

where Eq. (20) comes from the fact that $\Delta T_i \le T$ and some rewriting; Eq. (21) comes from the fact that $p = \lceil \log_2 T^\beta \rceil \ge \log_2 T^\beta$.

Combining Eq. (21) and Eq. (13) gives the following regret bound for iterations $i > i_\star$:

$$R_{\Delta T_i} \le 250 \, (\log_2 T)^{3/2} \cdot T^{\max\{\beta, 1+\alpha-\beta\}},$$

where the constant 250 simply comes from $84 + 166$.

Since the cumulative regret is non-decreasing in $t$, we have

$$R_T \le \sum_{i=1}^{p} R_{\Delta T_i}$$
$$\le 250 \, p \, (\log_2 T)^{3/2} \cdot T^{\max\{\beta, 1+\alpha-\beta\}}$$
$$\le 250 \, (\log_2 T + 1) \cdot (\log_2 T)^{3/2} \cdot T^{\max\{\beta, 1+\alpha-\beta\}} \tag{22}$$
$$\le 251 \, (\log_2 T)^{5/2} \cdot T^{\max\{\beta, 1+\alpha-\beta\}},$$

where Eq. (22) comes from the fact that $p = \lceil \log_2(T^\beta) \rceil \le \log_2(T^\beta) + 1 \le \log_2 T + 1$. Our results follows after noticing $R_T \le T$ is a trivial upper bound. $\square$

## A.2 Anytime version

---

**Algorithm 4:** Anytime version of `MOSS++`

---

**Input:** User specified parameter $\beta \in [1/2, 1]$.

  1: **for** $i = 0, 1, \dots$ **do**

  2:     Run Algorithm 1 with parameter $\beta$ for $2^i$ rounds (note that we will set
$p = \lceil \log_2 2^{i\beta} \rceil = \lceil i\beta \rceil$).

  3: **end for**

---

**Corollary 1.** *For any unknown time horizon $T$, run Algorithm 4 with an user-specified parameter $\beta \in [1/2, 1]$ leads to the following regret upper bound:*

$$\sup_{\omega \in \mathcal{H}_T(\alpha)} R_T \leq C \left(\log_2 T\right)^{5/2} \cdot T^{\min\{\max\{\beta, 1+\alpha-\beta\}, 1\}},$$

*where $C$ is a universal constant.*

*Proof.* Let $t_\star$ be the smallest integer such that

$$\sum_{i=0}^{t_\star} 2^i = 2^{t_\star+1} - 1 \geq T.$$

We then only need to run Algorithm 1 for at most $t_\star$ times. By the definition of $t_\star$, we also know that $2^{t_\star} \leq T$, which leads to $t_\star \leq \log_2 T$.

Let $\gamma = \min\{\max\{\beta, 1+\alpha-\beta\}, 1\}$. From Theorem 1 we know that the regret at $i \in [t_\star]$-th round, denoted as $R_{2^i}$, could be upper bounded by

$$R_{2^i} \leq 251 \left(\log_2 2^i\right)^{5/2} \cdot (2^i)^\gamma = 251\, i^{5/2} \cdot (2^\gamma)^i \leq 251\, t_\star^{5/2} \cdot (2^\gamma)^i \leq 251 \left(\log_2 T\right)^{5/2} \cdot (2^\gamma)^i.$$

For $i = 0$, we have $R_{2^0} \leq 1 \leq 251 \left(\log_2 T\right)^{5/2} \cdot (2^\gamma)^0$ as well as long as $T \geq 2$.

Now for the unknown time horizon $T$, we could upper bound the regret by

$$
\begin{aligned}
R_T &\leq \sum_{i=0}^{t_\star} R_{2^i} \\
&\leq 251 \left(\log_2 T\right)^{5/2} \cdot \left(\sum_{i=0}^{t_\star} (2^\gamma)^i\right) \\
&\leq 251 \left(\log_2 T\right)^{5/2} \cdot \int_{x=0}^{t_\star+1} (2^\gamma)^x dx \qquad\qquad (23) \\
&= 251 \left(\log_2 T\right)^{5/2} \cdot \frac{1}{\log 2^\gamma} \cdot \left((2^\gamma)^{t_\star+1} - 1\right) \\
&\leq \frac{2^\gamma}{\gamma \log 2}\, 251 \left(\log_2 T\right)^{5/2} \cdot T^\gamma \\
&\leq 1449 \left(\log_2 T\right)^{5/2} \cdot T^\gamma, \qquad\qquad\qquad\qquad (24)
\end{aligned}
$$

where Eq. (23) comes from upper bounding summation by integral; and Eq. (24) comes from a trivial bound on the constant when $1/2 \leq \gamma \leq 1$. $\qquad\qquad\qquad\qquad\qquad\qquad\qquad$ □

# B Omitted proofs for Section 4

## B.1 Proof of Theorem 2

**Theorem 2.** *For any $0 \leq \alpha' < \alpha \leq 1$, assume $T^\alpha \leq B$ and $\lfloor T^\alpha \rfloor - 1 \geq \max\{T^\alpha/4, 2\}$. If an algorithm is such that $\sup_{\omega \in \mathcal{H}_T(\alpha')} R_T \leq B$, then the regret of this algorithm is lower bounded on $\mathcal{H}_T(\alpha)$:*

$$\sup_{\omega \in \mathcal{H}_T(\alpha)} R_T \geq 2^{-10} T^{1+\alpha} B^{-1}. \qquad\qquad\qquad\qquad (2)$$

The proof of Theorem 2 is mainly inspired by the proofs of lower bounds in [23, 17]. Before the start of the proof, we first state a generalized version of Pinsker's inequality developed in [17] (Lemma 3 therein).

**Lemma 3.** *Let $\mathbb{P}$ and $\mathbb{Q}$ be two probability measures. For any random variable $Z \in [0, 1]$, we have*

$$|\mathbb{E}_{\mathbb{P}}[Z] - \mathbb{E}_{\mathbb{Q}}[Z]| \leq \sqrt{\mathrm{KL}(\mathbb{P}, \mathbb{Q})/2}.$$

We consider $K + 1$ bandit instances $\{\underline{\nu}_i\}_{i=0}^{K}$ such that each bandit instance is a collection of $n$ distributions $\underline{\nu}_i = (\nu_{i1}, \nu_{i2}, \ldots, \nu_{in})$ where each $\nu_{ij}$ represents a Gaussian distribution $\mathcal{N}(\mu_{ij}, 1/4)$ with $\mu_{ij} = \mathbb{E}[\nu_{ij}]$. For any given $0 \leq \alpha' < \alpha \leq 1$ and time horizon $T$ large enough, we choose $n, m_0, m, K \in \mathbb{N}_+$ such that the following three conditions are satisfied:

1. $n = m_0 + Km$;
2. $n/m_0 \leq 2T^{\alpha'}$;
3. $n/m \leq 2T^{\alpha}$.

**Proposition 1.** *Integers satisfying the above three conditions exist. For instance, we could first fix $m \in \mathbb{N}_+$ and set $K = \lfloor T^{\alpha} \rfloor - 1 \geq 2$.[8] One could then set $m_0 = m\lceil T^{\alpha-\alpha'} \rceil$ and $n = m_0 + Km$.*

*Proof.* We notice that the first condition holds by construction. We now show that the second and the third conditions hold.

For the second condition, we have

$$\begin{aligned}
\frac{n}{m_0} &= \frac{m_0 + Km}{m_0} \\
&= 1 + \frac{m(\lfloor T^{\alpha} \rfloor - 1)}{m \lceil T^{\alpha-\alpha'} \rceil} \\
&\leq 1 + \frac{T^{\alpha}}{T^{\alpha-\alpha'}} \\
&\leq 2T^{\alpha'}.
\end{aligned}$$

For the third condition, we have

$$\begin{aligned}
\frac{n}{m} &= \frac{m_0 + Km}{m} \\
&= \frac{m\lceil T^{\alpha-\alpha'} \rceil + (\lfloor T^{\alpha} \rfloor - 1)m}{m} \\
&= \lceil T^{\alpha-\alpha'} \rceil + \lfloor T^{\alpha} \rfloor - 1 \\
&= \left( \lceil T^{\alpha-\alpha'} \rceil - 1 \right) + \lfloor T^{\alpha} \rfloor \\
&\leq T^{\alpha-\alpha'} + T^{\alpha} \\
&\leq 2T^{\alpha}.
\end{aligned}$$

$\square$

Now we group $n$ distribution into $K + 1$ different groups based on their indices: $S_0 = [m_0]$ and $S_i = [m_0 + i \cdot m] \backslash [m_0 + (i-1) \cdot m]$. Let $\Delta \in (0, 1]$ be a parameter to be tuned later, we then define $K + 1$ bandit instances $\underline{\nu}_i$ for $i \in \{0\} \cup [K]$ by assigning different values to their means $\mu_{ij}$:

$$\mu_{ij} = \begin{cases} \Delta/2 & \text{if } j \in S_0, \\ \Delta & \text{if } j \in S_i \text{ and } i \neq 0, \\ 0 & \text{otherwise.} \end{cases} \tag{25}$$

We could clearly see there are $m_0$ best arms in instance $\underline{\nu}_0$ and $m$ best arms in instances $\underline{\nu}_i, \forall i \in [K]$. Based on our construction in Proposition 1, we could then conclude that, with time horizon $T$, the

regret minimization problem with respect to $\underline{\nu}_0$ is in $\mathcal{H}_T(\alpha')$; and similarly the regret minimization problem with respect to $\underline{\nu}_i$ is in $\mathcal{H}_T(\alpha)$, $\forall i \in [K]$.

For any $t \in [T]$, the tuple of random variables $H_t = (A_1, X_1, \ldots, A_t, X_t)$ is the outcome of an algorithm interacting with an bandit instance up to time $t$. Let $\Omega_t = ([n] \times \mathbb{R})^t \subseteq \mathbb{R}^{2t}$ and $\mathcal{F}_t = \mathfrak{B}(\Omega_t)$; one could then define a measurable space $(\Omega_t, \mathcal{F}_t)$ for $H_t$. The random variables $A_1, X_1, \ldots, A_t, X_t$ that make up the outcome are defined by their coordinate projections:

$$A_t(a_1, x_1, \ldots, a_t, x_t) = a_t \quad \text{and} \quad X_t(a_1, x_1, \ldots, a_t, x_t) = x_t.$$

For any fixed algorithm/policy $\pi$ and bandit instance $\underline{\nu}_i$, $\forall i \in \{0\} \cup [K]$, we are now constructing a probability measure $\mathbb{P}_{i,t}$ over $(\Omega_t, \mathcal{F}_t)$. Note that a policy $\pi$ is a sequence $(\pi_t)_{t=1}^T$, where $\pi_t$ is a probability kernel from $(\Omega_{t-1}, \mathcal{F}_{t-1})$ to $([n], 2^{[n]})$. For each $i$, we define another probability kernel $p_{i,t}$ from $(\Omega_{t-1} \times [n], \mathcal{F}_{t-1} \otimes 2^{[n]})$ to $(\mathbb{R}, \mathfrak{B}(\mathbb{R}))$ that models the reward. Assuming the reward is distributed according to $\mathcal{N}(\mu_{ia_t}, 1/4)$, we give its explicit expression for any $B \in \mathfrak{B}(\mathbb{R})$ as:

$$p_{i,t}\big((a_1, x_1, \ldots, a_t), B\big) = \int_B \sqrt{\frac{2}{\pi}} \exp\big(-2(x - \mu_{ia_t})\big) dx.$$

The probability measure over $\mathbb{P}_{i,t}$ over $(\Omega_t, \mathcal{F}_t)$ could then be define recursively as $\mathbb{P}_{i,t} = p_{i,t}\big(\pi_t \mathbb{P}_{i,t-1}\big)$. We use $\mathbb{E}_i$ to denote the expectation taken with respect to $\mathbb{P}_{i,T}$. Apply the same analysis as on page 21 of [17], we obtain the following proposition on KL decomposition.

**Proposition 2.**

$$\text{KL}\left(\mathbb{P}_{0,T}, \mathbb{P}_{i,T}\right) = \mathbb{E}_0\left[\sum_{t=1}^T \text{KL}\left(\mathcal{N}(\mu_{0A_t}, 1/4), \mathcal{N}(\mu_{iA_t}, 1/4)\right)\right].$$

With respect to notations and constructions described above, we now prove Theorem 2.

*Proof.* (Theorem 2) Let $N_{S_i}(T) = \sum_{t=1}^T \mathbb{1}\left(A_t \in S_i\right)$ denote the number of times the algorithm $\pi$ selects an arm in $S_i$ up to time $T$. Let $R_{i,T}$ denote the expected (pseudo) regret achieved by the algorithm $\pi$ interacting with the bandit instance $\underline{\nu}_i$. Based on the construction of bandit instance in Eq. (25), we have

$$R_{0,T} \geq \frac{\Delta}{2} \sum_{i=1}^K \mathbb{E}_0\left[N_{S_i}(T)\right], \tag{26}$$

and $\forall i \in [K]$,

$$R_{i,T} \geq \frac{\Delta}{2}\left(T - \mathbb{E}_i[N_{S_i}(T)]\right) = \frac{T\Delta}{2}\left(1 - \frac{\mathbb{E}_i[N_{S_i}(T)]}{T}\right). \tag{27}$$

According to Proposition 2 and the calculation of KL-divergence between two Gaussian distributions, we further have

$$\begin{aligned}
\text{KL}(\mathbb{P}_{0,T}, \mathbb{P}_{i,T}) &= \mathbb{E}_0\left[\sum_{t=1}^T \text{KL}\left(\mathcal{N}(\mu_{0A_t}, 1/4), \mathcal{N}(\mu_{iA_t}, 1/4)\right)\right] \\
&= \mathbb{E}_0\left[\sum_{t=1}^T 2\left(\mu_{0A_t} - \mu_{iA_t}\right)^2\right] \\
&= 2\,\mathbb{E}_0\left[N_{S_i}(T)\right]\Delta^2, \tag{28}
\end{aligned}$$

where Eq. (28) comes from the fact that $\mu_{0j}$ and $\mu_{ij}$ only differs for $j \in S_i$ and the difference is exactly $\Delta$.

We now consider the average regret over $i \in [K]$:

$$\frac{1}{K}\sum_{i=1}^{K} R_{i,T} \geq \frac{T\Delta}{2}\left(1 - \frac{1}{K}\sum_{i=1}^{K}\frac{\mathbb{E}_i[N_{S_i}(T)]}{T}\right)$$

$$\geq \frac{T\Delta}{2}\left(1 - \frac{1}{K}\sum_{i=1}^{K}\left(\frac{\mathbb{E}_0[N_{S_i}(T)]}{T} + \sqrt{\frac{\mathrm{KL}(\mathbb{P}_{0,T}, \mathbb{P}_{i,T})}{2}}\right)\right) \quad (29)$$

$$= \frac{T\Delta}{2}\left(1 - \frac{1}{K}\frac{\sum_{i=1}^{K}\mathbb{E}_0[N_{S_i}(T)]}{T} - \frac{1}{K}\sum_{i=1}^{K}\sqrt{\mathbb{E}_0\left[N_{S_i}(T)\right]\Delta^2}\right) \quad (30)$$

$$\geq \frac{T\Delta}{2}\left(1 - \frac{1}{K} - \sqrt{\frac{\sum_{i=1}^{K}\mathbb{E}_0\left[N_{S_i}(T)\right]\Delta^2}{K}}\right) \quad (31)$$

$$\geq \frac{T\Delta}{2}\left(1 - \frac{1}{K} - \sqrt{\frac{2\Delta R_{0,T}}{K}}\right) \quad (32)$$

$$\geq \frac{T\Delta}{2}\left(\frac{1}{2} - \sqrt{\frac{2\Delta B}{K}}\right), \quad (33)$$

where Eq. (29) comes from applying Lemma 3 with $Z = N_{S_i}(T)/T$ and $\mathbb{P} = \mathbb{P}_{0,T}$ and $\mathbb{Q} = \mathbb{P}_{i,T}$; Eq. (30) comes from applying Eq. (28); Eq. (31) comes from concavity of $\sqrt{\cdot}$ and the fact that $\sum_{i=1}^{K}\mathbb{E}_0[N_{S_i}(T)] \leq T$; Eq. (32) comes from applying Eq. (26); and finally Eq. (33) comes from the fact that $K \geq 2$ by construction and the assumption that $R_{0,T} \leq B$.

To obtain a large value for Eq. (33), one could maximize $\Delta$ while still make $\sqrt{2\Delta B/K} \leq 1/4$. Set $\Delta = 2^{-5}KB^{-1}$, following Eq. (33), we obtain

$$\frac{1}{K}\sum_{i=1}^{K} R_{i,T} \geq 2^{-8}TKB^{-1}$$

$$= 2^{-8}T\left(\lfloor T^\alpha \rfloor - 1\right)B^{-1} \quad (34)$$

$$\geq 2^{-10}T^{1+\alpha}B^{-1}, \quad (35)$$

where Eq. (34) comes from the construction of $K$; and Eq. (35) comes from the assumption that $\lfloor T^\alpha \rfloor - 1 \geq T^\alpha/4$.

Now we only need to make sure $\Delta = 2^{-5}KB^{-1} \leq 1$. Since we have $K = \lfloor T^\alpha \rfloor - 1 \leq T^\alpha$ by construction and $T^\alpha \leq B$ by assumption, we obtain $\Delta = 2^{-5}KB^{-1} \leq 2^{-5} < 1$ as desired. □

## B.2 Proof of Theorem 3

**Lemma 4.** *Suppose an algorithm achieves rate function $\theta$, then for any $0 < \alpha \leq \theta(0)$, we have*

$$\theta(\alpha) \geq 1 + \alpha - \theta(0). \quad (36)$$

*Proof.* Fix $0 < \alpha \leq \theta(0)$. For any $\epsilon > 0$, there exists constant $c_1$ and $c_2$ such that for sufficiently large $T$,

$$\sup_{\omega \in \mathcal{H}_T(0)} R_T \leq c_1 T^{\theta(0)+\epsilon} \quad \text{and} \quad \sup_{\omega \in \mathcal{H}_T(\alpha)} R_T \leq c_2 T^{\theta(\alpha)+\epsilon}.$$

Let $B = \max\{c_1, 1\} \cdot T^{\theta(0)+\epsilon}$, we could see that $T^\alpha \leq T^{\theta(0)} \leq B$ holds by assumption. For $T$ large enough, the condition $\lfloor T^\alpha \rfloor - 1 \geq \max\{T^\alpha/4, 2\}$ of Theorem 2 holds. We then have

$$c_2 T^{\theta(\alpha)+\epsilon} \geq 2^{-10}T^{1+\alpha}\left(\max\{c_1, 1\} \cdot T^{\theta(0)+\epsilon}\right)^{-1} = 2^{-10}T^{1+\alpha-\theta(0)-\epsilon}/\max\{c_1, 1\}.$$

For $T$ sufficiently large, we then must have

$$\theta(\alpha) + \epsilon \geq 1 + \alpha - \theta(0) - \epsilon.$$

Let $\epsilon \to 0$ leads to the desired result. □

**Lemma 5.** *Suppose an rate function $\theta$ is achieved by an algorithm, then we must have*

$$\theta(\alpha) \geq \min\{\max\{\theta(0), 1 + \alpha - \theta(0)\}, 1\}, \tag{37}$$

*with $\theta(0) \in [1/2, 1]$.*

*Proof.* For any rate function $\theta$ achieved by an algorithm, we first notice that $\theta(\alpha) \geq \theta(\alpha')$ for any $0 \leq \alpha' < \alpha \leq 1$ since $\mathcal{H}_T(\alpha') \subseteq \mathcal{H}_T(\alpha)$; this also implies $\theta(\alpha) \geq \theta(0)$. From Lemma 4, we further obtain $\theta(\alpha) \geq 1 + \alpha - \theta(0)$ if $\alpha \leq \theta(0)$. Thus, for any $\alpha \in (0, \theta(0)]$, we have

$$\theta(\alpha) \geq \max\{\theta(0), 1 + \alpha - \theta(0)\}. \tag{38}$$

Note that this indicates $\theta(\theta(0)) = 1$, as we trivially have $R_T \leq T$. For any $\alpha \in (\theta(0), 1]$, we have $\theta(\alpha) \geq \theta(\theta(0)) = 1$, which leads to $\theta(\alpha) = 1$ for $\alpha \in [\theta(0), 1]$. To summarize, we obtain the desired result in Eq. (37). We have $\theta(0) \in [1/2, 1]$ since the minimax optimal rate among problems in $\mathcal{H}_T(0)$ is $1/2$. $\square$

**Theorem 3.** *The rate function achieved by* `MOSS++` *with any $\beta \in [1/2, 1]$, i.e.,*

$$\theta_\beta : \alpha \mapsto \min\{\max\{\beta, 1 + \alpha - \beta\}, 1\}, \tag{3}$$

*is Pareto optimal.*

*Proof.* From Theorem 1, we know that the rate in Eq. (3) is achieved by Algorithm 1 with input $\beta$. We only need to prove that no other algorithms achieve strictly smaller rates in pointwise order.

Suppose, by contradiction, we have $\theta'$ achieved by an algorithm such that $\theta'(\alpha) \leq \theta_\beta(\alpha)$ for all $\alpha \in [0, 1]$ and $\theta'(\alpha_0) < \theta(\alpha_0)$ for at least one $\alpha_0 \in [0, 1]$. We then must have $\theta'(0) \leq \theta_\beta(0) = \beta$. We consider the following two exclusive cases.

**Case 1** $\theta'(0) = \beta$**.** According to Lemma 5, we must have $\theta' \geq \theta_\beta$, which leads to a contradiction.

**Case 2** $\theta'(0) = \beta' < \beta$**.** According Lemma 5, we must have $\theta' \geq \theta_{\beta'}$. However, $\theta_{\beta'}$ is not strictly better than $\theta_\beta$, e.g., $\theta_{\beta'}(2\beta - 1) = 2\beta - \beta' > \beta = \theta_\beta(2\beta - 1)$, which also leads to a contradiction. $\square$

## C  Omitted proofs for Section 5

### C.1  Proof of Lemma 1

**Lemma 1.** *Suppose $\alpha$ is the true hardness parameter and $\alpha_i - 1/\lceil \log T \rceil < \alpha \leq \alpha_i$, run Algorithm 2 with time horizon $T$ and $\alpha_i$ leads to the following regret bound:*

$$\sup_{\omega \in \mathcal{H}_T(\alpha)} R_T \leq C \log T \cdot T^{(1+\alpha)/2},$$

*where $C$ is a universal constant.*

*Proof.* According to Lemma 2, the definition of $\alpha$ and the assumption that $\alpha \leq \alpha_i$, we know that $\mathbb{P}(\mathcal{E}) \leq 1/\sqrt{T}$. We now upper bound the regret:

$$R_T \leq \left(39\sqrt{|S_{\alpha_i}|T} + |S_{\alpha_i}|\right) \cdot \mathbb{P}(\neg\mathcal{E}) + T \cdot \mathbb{P}(\mathcal{E}) \tag{39}$$

$$\leq \left(39\sqrt{|S_{\alpha_i}|T} + |S_{\alpha_i}|\right) \cdot 1 + T \cdot \frac{1}{\sqrt{T}}$$

$$\leq 56\,(\log T)^{1/2} \cdot T^{(1+\alpha_i)/2} + 2\log T \cdot T^{\alpha_i} + \sqrt{T}$$

$$\leq 59 \log T \cdot T^{(1+\alpha_i)/2}$$

$$< 59 \log T \cdot T^{(1+\alpha)/2} \cdot T^{1/(2\lceil \log T \rceil)} \tag{40}$$

$$\leq 59\sqrt{e} \log T \cdot T^{(1+\alpha)/2}, \tag{41}$$

where Eq. (39) comes from the regret bound of `MOSS`; Eq. (40) comes from the assumption that $\alpha_i < \alpha + 1/\lceil \log T \rceil$; and Eq. (41) comes from the fact that $T^{1/(2\lceil \log T \rceil)} = e^{(\log T/(2\lceil \log T \rceil))} \leq \sqrt{e}$.[9]

$\square$

## C.2 Proof of Theorem 4

We first provide a martingale (difference) concentration result from [29] (a rewrite of Theorem 2.19).

**Lemma 6.** *Let $\{D_t\}_{t=1}^{\infty}$ be a martingale difference sequence adapted to filtration $\{\mathcal{F}_t\}_{t=1}^{\infty}$. If $\mathbb{E}[\exp(\lambda D_t)|\mathcal{F}_{t-1}] \leq \exp(\lambda^2 \sigma^2/2)$ almost surely for any $\lambda \in \mathbb{R}$, we then have*

$$\mathbb{P}\left(\left|\sum_{i=1}^{t} D_i\right| \geq \epsilon\right) \leq 2\exp\left(-\frac{\epsilon^2}{2t\sigma^2}\right).$$

**Theorem 4.** *For any $\alpha \in [0,1]$ unknown to the learner, run* `Parallel` *with time horizon $T$ and optimal expected reward $\mu_\star$ leads to the following regret upper bound:*

$$\sup_{\omega \in \mathcal{H}_T(\alpha)} R_T \leq C\,(\log T)^2\,T^{(1+\alpha)/2},$$

*where $C$ is a universal constant.*

*Proof.* This proof largely follows the proof of Theorem 4 in [23]. For any $T \in \mathbb{N}_+$ and $i \in [\lceil \log T \rceil]$, recall $\mathtt{SR}_i$ is the subroutine initialized with $T$ and $\alpha_i = i/[\lceil \log T \rceil]$. We use $T_{i,t}$ to denote the number of samples allocated to $\mathtt{SR}_i$ up to time $t$, and represent its empirical regret at time $t$ as $\widehat{R}_{i,t} = T_{i,t} \cdot \mu_\star - \sum_{t=1}^{T_{i,t}} X_{i,t}$ where $X_{i,t} \sim \nu_{A_{i,t}}$ is the $t$-th empirical reward obtained *by* $\mathtt{SR}_i$ and $A_{i,t}$ is the index of the $t$-th arm pulled *by* $\mathtt{SR}_i$. We consider the corresponding regret $R_{i,t} = T_{i,t} \cdot \mu_\star - \sum_{t=1}^{T_{i,t}} \mathbb{E}[\mu_{A_{i,t}}]$ (which is random in $T_{i,t}$). We choose $\delta = 1/\sqrt{T}$ as the confidence parameter and provide $\delta' = \delta/\lceil \log T \rceil$ failure probability to each subroutine.

Notice that $R_{i,t} - \widehat{R}_{i,t} = \sum_{t=1}^{T_{i,t}} \left(X_{i,t} - \mathbb{E}[\mu_{A_{i,t}}]\right)$ is a martingale with respect to filtration $\mathcal{F}_t = \sigma\left(\bigcup_{i\in[\lceil \log T \rceil]}\{T_{i,1}, A_{i,1}, X_{i,1}, \dots, T_{i,t}, A_{i,T_{i,t}}, X_{i,T_{i,t}}\}\right)$; and $(R_{i,t} - \widehat{R}_{i,t}) - (R_{i,t-1} - \widehat{R}_{i,t-1})$ defines a martingale difference sequence. Since, no matter what value $T_{i,t}$ takes, $X_{i,T_{i,t}} - \mathbb{E}[\mu_{A_{i,T_{i,t}}}] = (X_{i,T_{i,t}} - \mu_{A_{i,T_{i,t}}}) + (\mu_{A_{i,T_{i,t}}} - \mathbb{E}[\mu_{A_{i,T_{i,t}}}])$ is $(\sqrt{2}/2)$-sub-Gaussian (following a similar analysis as in Eq. (7)), applying Lemma 6 together with a union bound gives:

$$\mathbb{P}\left(\forall i \in [\lceil \log T \rceil], \forall t \in [T] : |\widehat{R}_{i,t} - R_{i,t}| \geq \sqrt{T_{i,t} \cdot \log\left(2T\lceil \log T \rceil/\delta\right)}\right) \leq \delta. \quad (42)$$

We use $\mathcal{E} = \left\{\forall i \in [\lceil \log T \rceil], \forall t \in [T] : |\widehat{R}_{i,t} - R_{i,t}| < \sqrt{T_{i,t} \cdot \log\left(2T\lceil \log T \rceil/\delta\right)}\right\}$ to denote the good event that holds true with probability at least $1 - \delta$. Since the regret could be trivially upper bounded by $T \cdot \delta = \sqrt{T}$ when $\mathcal{E}$ doesn't hold, we only focus on the case when event $\mathcal{E}$ holds in the following.

Fix any subroutine $k \in [\lceil \log T \rceil]$ and consider its empirical regret $\widehat{R}_{k,T}$ up to time $T$. For any $j \neq k$, let $T_j \leq T$ be the last time that the subroutine $\mathtt{SR}_j$ was invoked, we have

$$\widehat{R}_{j,T_j} \leq \widehat{R}_{k,T_j}$$
$$\leq R_{k,T_j} + \sqrt{T_{k,T_j} \cdot \log\left(2T\lceil \log T \rceil/\delta\right)}$$
$$\leq R_{k,T} + \sqrt{T \cdot \log\left(2T\lceil \log T \rceil/\delta\right)}, \quad (43)$$

where Eq. (43) comes from the fact that the cumulative regret $R_{k,t}$ in non-decreasing in $t$. Since $\mathtt{SR}_j$ will only run additional $\lceil \sqrt{T} \rceil$ rounds after it was selected at time $T_j$, we further have

$$\widehat{R}_{j,T} \leq \widehat{R}_{j,T_j} + \lceil \sqrt{T} \rceil$$
$$\leq R_{k,T} + \sqrt{5T \cdot \log\left(2T\lceil \log T \rceil/\delta\right)}, \quad (44)$$

where Eq. (44) comes from the combining Eq. (43) with a trivial bounding $\lceil \sqrt{T} \rceil \leq \sqrt{4T}$ for all $T \in \mathbb{N}_+$. Combining Eq. (44) with the fact that $R_{j,T} \leq \widehat{R}_{j,T} + \sqrt{T \cdot \log\left(2T\lceil \log T \rceil/\delta\right)}$ leads to

$$R_{j,T} \leq R_{k,T} + 4\sqrt{T \cdot \log\left(2T\lceil \log T \rceil/\delta\right)}. \quad (45)$$

Let $i_\star \in [\lceil \log T \rceil]$ denote the index such that $\alpha_{i_\star - 1} < \alpha \le \alpha_{i_\star}$. As the total regret is the sum of all subroutines, we have that, for some universal constant $C$,

$$\sum_{i=1}^{\lceil \log T \rceil} R_{i,T} \le \lceil \log T \rceil \cdot \left( R_{i_\star, T} + 4\sqrt{T \cdot \log\left(2T\lceil \log T\rceil / \delta\right)} \right) \tag{46}$$

$$\le \lceil \log T \rceil \cdot \left( 59\sqrt{e} \, \log T \cdot T^{(1+\alpha)/2} + 4\sqrt{T \cdot \log\left(2T^{3/2}\lceil \log T\rceil\right)} \right) \tag{47}$$

$$\le C \, (\log T)^2 \, T^{(1+\alpha)/2},$$

where Eq. (46) comes from setting $k = i_\star$ in Eq. (45); Eq. (47) comes from applying Lemma 1 with the non-decreasing nature of cumulative regret and taking $\delta = 1/\sqrt{T}$. Integrate once more leads to the desired result. $\qquad\square$

## C.3 Anytime version

The anytime version of Algorithm 3 could be constructed as following.

---
**Algorithm 5:** Anytime version of `Parallel`

---
1: **for** $i = 0, 1, \ldots$ **do**
2:      Run Algorithm 3 with the optimal expected reward $\mu_\star$ for $2^i$ rounds.
3: **end for**

---

**Corollary 2.** *For any time horizon $T$ and $\alpha \in [0, 1]$ unknown to the learner, run Algorithm 5 with optimal expected reward $\mu_\star$ leads to the following anytime regret upper:*

$$\sup_{\omega \in \mathcal{H}_T(\alpha)} R_T \le C \, (\log T)^2 \, T^{(1+\alpha)/2},$$

*where $C$ is a universal constant.*

*Proof.* The proof is similar to the one for Corollary 1. $\qquad\square$