[Reviews · NeurIPS 2020]

Review 1

Summary and Contributions: The paper studies cumulative regret when there are multiple best arms and the number of best arms is unknown. It defines a new notion of difficulty for such problems, \psi. It provides a new algorithm for this problem that is Pareto optimal over problems with different hardness levels. It also proves that there is no algorithm that is simultaneously optimal across all problem difficulties. However, the authors show that if mu_* is known then simultaneous optimality is possible. Finally, they demonstrate the superior performance of their algorithms in experiments.

Strengths: The regime where n can be larger than T is important and well-motivated by many modern appplications. Their observation that there are various levels of difficulty for this problem and it is not possible to be simultaneously optimal for all of them is very insightful. It seems that Theorem 2 implies that no algorithm is within log(T) of the minimax rate for all problem difficulties, correct? If so this polynomial gap seems quite convincing. Algorithm 1 is a non-trivial algorithm and leads to an algorithm with strong empirical performance. The experiments are thorough.

Weaknesses: The paper does not seem to develop fundamentally new algorithmic ideas, although they apply sophisticated algorithms from the literature to an important problem and in an enlightening way. It would seem that mu_* is never known exactly in practice. How does misspecification of mu_* affect the performance of algorithm 3? is Minimax optimality achievable even under misspecification? How would this affect empirical performance?  It seems odd that Algorithm 3 does worse in the experiments even though it has extra information. Is there a way to get around the issue that you are running several versions of the algorithm? Does it remain an open question how to leverage this extra information into a empirically superior algorithm?

Correctness: Yes.

Clarity: The paper is extremely clear.

Relation to Prior Work: Yes.

Reproducibility: Yes

Additional Feedback: After rebuttal: I have read the response and other reviews and my score remains the same.


Review 2

Summary and Contributions: The authors consider a stochastic bandit setting with n arms, where n is taken to be very large. Unlike other large-armed bandit work, there is no structure assumed between the payoff of the various arms. Their methods rely heavily on MOSS [Audibert et al.], which attains an optimal regret rate of \sqrt{nT} in the usual setting where n <= T. Letting m <= n denote the number of arms that are optimal (or near optimal), the authors begin by defining alpha satisfying n/m = T^{\alpha} as a key quantity characterizing the hardness. They observe that if alpha were known, one could sample a subset of the arms of cardinality T^alpha log T, which contains an optimal arm whp. Applying MOSS attains a regret bound of O(T^(1+alpha)/2). Meanwhile, there exists an instance with parameter alpha, and matching lower bound (when m = 1 and n = T^alpha). Thus, the key difficulty is attaining this same result when alpha is not known. They give an algorithm that runs multiple epochs of MOSS for exponentially increasing rounds on subsampled arms. In each epoch, the number of sampled arms decays by a factor of 2. However, the algorithm also includes a “virtual” arm that simulates the empirical distribution of arms played on each of the previous epochs. The algorithm’s analysis is very clean and follows by identifying an epoch i* where the number of samples arms falls below T^{alpha} \log T. Before i*, so many arms are sampled, that an optimal one is bound to be among them. After i*, the performance of the virtual arm becomes near-optimal. The ultimate bound, however, depends on a correct setting of a user-specified parameter beta in order to achieve the desired rate of O(T^(1+alpha)/2). The remainder of the theoretical content of the paper essentially defends this property of their algorithm in two ways. (1) They argue that no algorithm can be optimal for all levels of alpha, and that their algorithm sits on the Pareto frontier, while naive algorithms such as subsampling arms and running MOSS do not, and (2) with knowledge of the mean of the best arm, an algorithm achieving the lower bound is possible. Finally, they give experiments demonstrating the performance of their algorithms at at fixed beta, and varying alphas. They also give an algorithm inspired by the theory but that is more empirically robust (re-using statistics from previous epochs). The experiments demonstrate strong performance until alpha becomes larger than user-defined beta (at which point the theory predicts vacuous regret bounds).

Strengths: 1) The paper does a very good job of analyzing a large MAB problem without a great deal of cumbersome assumptions. The only salient quantity is the fraction of arms that are optimal. 2) The paper is very clear and well written. 3) The algorithmic ideas and analysis are interesting, and could be of value beyond the setting considered.

Weaknesses: 1) The optimality of the bound ultimately depends on a hyperparameter being set properly. However, the authors do a good job defending this, and one can imagine making reasonable choices for this hyperparameter in practice.

Correctness: The proofs appear to be correct, and the empirical methodology sound.

Clarity: The paper is easy to read.

Relation to Prior Work: The authors do a good job of setting the material in context of previous work. I don't think there is missing literature except for more large-arm/continuum bandit work, which is only tangentially related due to the lack of smoothness assumptions here.

Reproducibility: Yes

Additional Feedback: 115: Definition of psi. The setup claims that n can be infinite, but then it seems that this definition would cause problems… Can the authors clarify? 137: typo “an” 138: I think it is standard enough to not warrant definition, but take note that [K] notation is used before it is defined (e.g. first paragraph in section 2). 167: typo -> “still holds” Appendix: 427: What does “valid algorithm” mean in this context? If anything, it seems like you would want to upper bound T_p (i.e., show that the algo does not exceed its time horizon). 446: A is used as the random arm, so probably better to call this event something else. 450: Typo: “if it exists” 454: Typo: “particular”


Review 3

Summary and Contributions: The authors consider the regret minimization problem with the existence of multiple best arms in the multi-armed bandit setting. Precisely, given a bandit problem with n arms, m optimal arms and horizon T, they define the hardness of this problem as \alpha = \log(n/m) /\log(T). They show that the minimax rate over the class of bandit problems such that their hardness is lower than a fixed \alpha is of order T^{(1+\alpha)/2}. Then they propose algorithm Restarting that without the knowledge of \alpha enjoys a regret of order T^{min(\max(\beta,1+\alpha-\beta),1)} for a bandit problem of hardness at most \alpha ad where $\beta\in[1/2,1] is some parameter of the algorithm. They also prove that it is not possible to construct an algorithm that is simultaneously optimal for all the classes of bandit problems of hardness at most \alpha. Nevertheless, they show that Restarting is Pareto optimal. When the \mean of an optimal arm is known they propose algorithm Parallel which matches the minimax rate simultaneously for all \alpha. Finally, they compare empirically Restarting, Parallel with MOSS algorithm, MOSS tuned knowing the hardness \alpha and Quantile an algorithm proposed by Chaudhuri and Kalyanakrishnan (2018).

Strengths: -methodological: notion of hardness (significance: medium). -theoretical:minimax rate for the class of bandit problems with a common upper bound on the hardness (significance: low), impossibility to adapt to the hardness of a problem (Th3) (significance: medium). -algorithmic: algorithm Restarting Pareto optimal and Parallel with the minimax rate O(T^{(1+\alpha)/2}) for all \alpha (with the knowledge of \mu^\star). (significance: medium).

Weaknesses: The impossibility to adapt to the hardness is interesting. For Th 4, the extra information about the optimal mean probably change also the lower bound (see Th 3), thus it is difficult to see how sharp this bound is. From a technical point of view, as acknowledged by the authors, algorithm Restarting borrows ideas from [17] and Parallele from [23]. I could increase my score if the few doubts on the proofs are cleared up.

Correctness: The proofs seem correct except few points in the proof of Theorem 2 and Theorem 3 (see specific comments below).

Clarity: The presentation is clear (see specific comments below).

Relation to Prior Work: It is clearly discussed. It could be interesting to elaborate on the link between your setting/class of bandits problems and the infinitely-armed bandit problem [6,30,27]. In particular is there any link between the hardness \alpha and the parameter they used to characterize the probability to sample an arm with a mean close to the optimal mean.

Reproducibility: Yes

Additional Feedback: Specific comments: L97: In the case where n=+\infty the max could not be defined and if we replace it by the sup S_\star could be empty. L108: maybe largely neglected is a bit too strong, for example when the regret is studied from a minimax point of view T is a part of the problem instance. L115: In fact \alpha = \log(n/m)/log(T)? L116: It is really counter-intuitive to set T^0= c\geq 4 and arbitrary. I do not understand why it is important to avoid the case with all best arms. Maybe it could be better to add a multiplicative constant: \inf{\alpha : n/m \leq 4 T^\alpha} L143: precise what you mean by randomly selected. L145: define clearly the mixture arm (with a formula). L191: Explain a bit why this result implies that it is impossible to achieve simultaneously a minimax regret for all \alpha. L218: It is not really an optimal strategy since its requires extra-information. L243: The extra information about the optimal mean probably change also the lower bound (see Th 3), thus it is difficult to see how sharp this bound is. What would be the minimal additional assumption such that adaptivity is possible? L450: can you explain why \Delta T_i \leq T? L455: It is 153 in the first inequality and you need to assume that T\geq 2. L463: I still not understand why \DeltaT_i \leq T because for i=p: \DeltaT_i = 2^{2p} \geq T^{2\beta} > T if \beta >1/2. L539: (28) there is a missing factor 2 if we follow (26) but this factor 2 in (26) is not necessary. L545: why this 2^{-5} in the definition of \Delta L559: max(c_1,1) L566: this part of the proof is not clear at all. Is \theta such that \theta \leq \theta_\beta ? Why can we suppose that \theta(0) = \beta ? ----Post rebuttal---- I have read the other reviews and the rebuttal. The authors addressed my concerns. I change my score accordingly.


Review 4

Summary and Contributions: This paper studies the optimal strategies when the number of arms is larger than the time horizon. The paper provides an algorithm that is able to pull a large number of arms given the time horizon, with an assumed density (beta). Whether the algorithm yields any optimal arms depends on the true density (alpha) of the optimal arms. The paper also argues that the provided algorithm is minimax optimal if alpha<beta yet fundamental hardness exists if alpha is large.

Strengths: * Theoretical grounding: The paper studies a basic question and provides some sound arguments. I did not follow everything but the cases where I spot-checked made sense. * Novelty of contribution: While there is prior work on pulling from infinite pool of arms (QRM), this work is unique by providing an adaptive minimax-optimal rate under fewer assumptions.

Weaknesses: * Practicality: the algorithmic improvements seem trivial. It is just an MOSS algorithm with some micro improvements based on law of iterated logarithm. While there is an EMP version to fine-tune the parameters, I am not seeing how it introduces practical changes if we similarly fine-tune MOSS. * Self-containment: The paper omitted the explanation of MOSS (through reference, I guess this method is equivalent to EXP4, which is better known).

Correctness: The theorems and intuitions make sense. I spot-checked Algorithm 1 with some numeric examples, which I hope to be included in the paper (see detailed suggestions). It might be worthwhile to see the counter example in Theorem 2, but I have not done that yet.

Clarity: Ok for a technical audience, though there may be some concerns about practical impacts which should be better discussed at the beginning of the paper.

Relation to Prior Work: Yes. The algorithm seems also related to Distilled Sensing from Rob Nowak's group.

Reproducibility: Yes

Additional Feedback: Line 96. what is the reason for 1/4-sub-Gaussian? Line 116-121, why not combine both equations into one? The equations are not numbered. Algorithm 1. The numeric example: set beta=1/2, T=2^20 (1million). this yields p=10, K1=2^11, K2=2^10, ... T1=2^11, T2=2^12, ... So, in the first iteration, more than half of them items are included. This yields some connections to distilled sensing. Figure 2. I am glad about the reference to QRM2, which is a relevant and memorable work. However, I don't have a color printer so I could only assume that this paper did better. What is the difference between Restarting and RestartingEmp? Why is RestartingEmp not discussed in the method section?


Review 5

Summary and Contributions: The paper studies bandit instances where the number of arms is larger than the horizon. Usual asymptotic analyses of the regret are typically of no use in this case. The authors focus on cases where there are multiple optimal arms and define a complexity measure that relies on the proportion of optimal arms and on the horizon. Namely, the complexity of a bandit instance with $n$ arms, $m$ best arms and a horizon $T$ as $\inf_{\alpha \in [0, 1]} {n/m \leq T^{\alpha}}$. They propose an algorithm that is adaptive to the complexity defined like so. They provide an upper bound of the regret, scaling like a power of $T$, with an exponent being a function of the parameter of the algorithm. The optimal choice of the parameter yields a regret of the order of $T^{(1+ \alpha}/2}$, which is the non-adaptive-minimax-optimal rate. They derive a lower bound and prove that the algorithm is Pareto optimal. Next, they provide another algorithm for the case when the reward of the best arms is known in advance. The analysis of the regret shows that the regret is non-adaptive-minimax-optimal. Some experiments illustrate these findings but fail to show the superiority of the second algorithm (which has access to additional information) over the first one in practice.

Strengths: I appreciate the originality of the paper, that comes from the fact that it introduces a measure of complexity, which I believe is new, and which relies on the proportion of optimal arms and on the horizon, while the usual measures of complexity usually omit to take the horizon into account. The results include a nice lower bound that shows that it is impossible to construct an algorithm that achieves minimax optimality for all complexities simultaneously. Despite this impossibility, the authors build an adaptive algorithm that is Pareto optimal. The presence of experiments, which show the behaviour of the regret with respect to the complexity and the time is also a positive element.

Weaknesses: - The fact that the measure of the complexity relies on the number of optimal arms is rather questionable, since there are not many applications where there can be a large set of optimal arms. I think the paper would benefit from a longer discussion of the generalization to near-optimal arms. Also, there does not seem to be any result of this kind in the Appendix, which is regrettable. - The lower bound in section 2, on which the hardness measure is based, relies on an example where there is a single best arm. At this point in the paper, it would be interesting to know of a lower bound for a case with more arms, in order to know if the hardness classes are uniform in this regard (i.e. if the claim that “problems with different time horizons but the same $\alpha$ are equally difficult in terms of the achievable minimax regret (the exponent of $T$) is true). Later on, we learn that a regret of the order of $T^{1+\alpha}$ can be achieved over the whole class, thanks to the restart algorithm, but it does not fully answer the question. - In Section 3, the authors rightly point out that the provided upper bound on the regret does not give any guarantees when $\beta<\alpha$ since it boils down to saying that the regret is bounded by $T$. The impossibility to achieve the non-adaptive minimax regret bound for every hardness class simultaneously does not mean that algorithms are bound to be this inefficient on a large range of $alpha$s. Furthermore, the choice of $\beta = 0.5$ in the experiments puts us exactly in the situation where $\beta<\alpha$ for half of the choices of $\alpha$. This raises two questions : -Can the bound be improved (the experiments seem to indicate that it can)? -What choice of $\beta$ should a user make when agnostic of $\alpha$ ? An answer or a discussion about these questions would have been appreciated. Even a graph showing the influence of the parameter on the regret would have been useful. - Although I understand that the theoretical result that additional information about the value of the best arm allows to achieve minimax optimality is satisfactory, I wonder whether this case is really relevant. I do not know of any practical case where the value of the optimal arm would be known in advance. - Another slightly weak point of the paper is that the experiments have been made with only 100 Monte Carlo trials for a horizon of 50,000 time steps. - Lastly, the paper seems to have been hastily written, which makes it difficult to read due to the large number of typos. I have read the authors' response and agree on their comment on the first and second points. I think that the answer to point 2 should figure in the paper, for it to be complete. On the role of $\beta$, I can only believe the authors, since I can not see the results of the experiments.

Correctness: Most probably, yes, but there are some greys areas, which I reported in the above paragraph.

Clarity: The structure of the paper is fairly clear but there is an important number of typos. I would recommend that it be proof-read, at least for grammar issues.

Relation to Prior Work: The structure of the paper is fairly clear but there is an important number of typos. I would recommend that it be proof-read, at least for grammar issues. The explanation of the proof in Section 3.1 is hard to understand without having read the whole proof. Rewriting this part a little would maybe help.

Reproducibility: Yes

Additional Feedback: # Some typos in the Appendix: R.427: $T_i=\sum_{j=1}^i T_i $ The last subscript is a $j$. R.428: I computed $T_p= 2^{p+1}(2^{p} -1) \geq 2^{2p} \geq T^{2 \beta}$ but I do not understand the current justification.The authors should indicate what a valid algorithm is. Also, I think that it is implicit that the algorithm stops as soon as it reaches $T$. I think this should be made explicit in the main body. R.437: The sentence “Thanks to the expectation on $A_t$ , this leads to the same result as if we also bring [...] into the analysis” is difficult to understand. R.440: This is not a typo, but the authors fail to explain where the 153 factor comes from. # Some of the grammar typos in the main body: R.1: “We study [a] regret minimization problem” R.11: “there is no algorithm [that] can be optimal” R.45 : “In Section 2 [we] formally define [the] regret minimization problem” R.46: ”the tension between [a] very large action space and [a] limited time horizon” R.49: “that indicates [that] there is no algorithm [that] can be” R.70: “ an anytime algorithm works under the same assumption.” -> “ an anytime algorithm that works under the same assumption.” R.92: “an additional logarithmic factors” -> ”additional logarithmic factors” R.96: “([that]could be infinite)” R.100: “T dependence” -> “dependent on T” R.108: “that T being part of the problem instance” -> “that T is part of the problem instance” R.109: “interested in the case [when n is comparable]” R.146: “i.e., the $k$-th element of $\hat{p}_i$ is [the] number of times” R.154: “is in reminiscent” -> “is reminiscent” R.167: “still hold[s]” R.215: ”and allow time horizon dependent quantile” -> “and allowing time horizon dependent quantile” R.223: ”instances of [a] subroutine” R.231: “[the] subroutine with the lowest”, “(break[ing] tie[s] arbitrarily)”, “and resume[s]” R. 239: “allocates new samples to [the] subroutine” R.240: “in the same order” -> “of the same order” R.242: “Parallel then achieve[s]” R.248: “how [the] regret curve” R.253: “with respect to mean at the (1 − $\rho$)-th quantile” -> ? R.257: “to represents” -> “to represent” R.269: “due to [the] large action space and [the] relatively small time horizon.” R.278: “MOSS Oracle need[s]” R.279: “guarantee the finding [of] at least one best arm” R.281: “We compare how [the] regret curve” R.286: “Restarting and Quantile doesn’t flat” -> “The regret curves of Restarting and Quantile do not flatten” R.300: “Using these Bernoulli reward models, we compare the performance of each algorithm is shown in Fig. 3.” -> ? R.309: “it’s cumulative regret” -> “its cumulative regret” R.325: “that could adapt the unknown hardness level” -> “ “that could adapt to the unknown hardness level” R338: “as an practical version” -> “as a practical version”

[Author Response · NeurIPS 2020]

We thank reviewers for appreciating the originality of our work and providing constructive feedback. All typos and
grammatical mistakes will be corrected in the final version. We address specific concerns below.

**Review 1: 1.** Yes, no algorithm can be minimax optimal for all $\alpha$ without additional assumptions. **2.** Minimax
optimality could still be achieved by Alg. 3 with $\mu_\star$ being mis-specified up to error $O(1/\sqrt{T})$; and similar empirical
performance is obtained under mis-specification. **3.** Maintaining $O(\log T)$ subroutines hurts the empirical performance
of Alg. 3. Designing an empirically superior algorithm that uses knowledge of $\mu_\star$ remains an open question.

**Review 2: 1.** For any chosen hyper-parameter $\beta \in [1/2, 1]$, Alg. 1 is Pareto optimal, and no algorithm can be strictly
better in terms of adaptivity. **2.** Our setting can be generalized to case with $n$ being infinite with a bit care (thanks for
pointing this out): in the infinite arm setting, $m$ is infinite as well (what matters is the ratio $n/m$: one can consider em-
bedding arms into $[0, 1]$ with the set of best arms having positive measure, but *without* additional structure assumptions).
**3.** The algorithm is *valid* in the sense that it selects an arm $A_t$ for any $t \in [T]$, i.e., it does not terminate before time $T$.

**Review 3: 1.** If $n$ is infinite, then there are indeed cases where the maximum is undefined. To avoid the potential
problem of empty $S_\star$, we advocate defining $S_\star$ in terms of $\epsilon$-good arms. **2.** Although $T$ was incorporated in lower
bounds, to the best of our knowledge, we are the first to incorporate $T$ into the cumulative regret minimization problem
$\mathcal{R}(n, m, T)$, and quantify corresponding hardness level $\alpha$. Previous work [4, 30] developed hardness parameters in
terms of the reservoir distribution of arms ($T$ not included; they additionally require those parameters to be *known*)
and thus cannot be directly related to $\alpha$. **3.** We defined $\psi$ as in Section 2 to avoid the trivial case with all best arms:
any algorithm is optimal in such case and achieves 0 regret. **4.** Random selection in Alg. 1 means sampling *uniformly*
*at random without replacement*. **5.** The virtual arm could be mathematically defined as $\widetilde{\nu}_i = \sum_{j=1}^n \widehat{p}_i(j) \cdot \nu_j$, where
$\widehat{p}_i(j)$ denotes the $j$-th element of the empirical sampling frequency $\widehat{p}_i$. **6.** The intuition behind Thm. 2 in explained
in the paragraph above it. But to interpret Thm. 2 alone: for any algorithm considered, if $B$ (or more precisely
$\sup_{\omega \in \mathcal{H}_T(\alpha')} R_T$) is large, we directly know that the algorithm is not optimal on the easy problem within $\mathcal{H}_T(\alpha')$;
if $B$ is small, the RHS of Eq.(2) is large and then the algorithm cannot be optimal on the hard problem within $\mathcal{H}_T(\alpha)$.
**7.** Whether sharper lower bounds are possible when given extra information about the optimal mean value is an open
problem, as is the question of the minimal additional assumption/information needed to fully adapt to $\alpha$. All we know
is no algorithm is simultaneously minimax optimal for all values of $\alpha$ without additional assumptions, and that given
the optimal mean value it is possible to be more adaptive to $\alpha$ than without it. **8.** We use $\Delta T_i$ to represent the length
of the $i$-th iteration *within* the total horizon $T$, and it really should have been defined as $\Delta T_i = \min\{2^{p+i}, T\}$ so that
$\Delta T_i \leq T$ always holds. **9.** We assume $T \geq 2$ on line 139. **10.** There is no missing factor of 2 in Eq.(28) and Eq.(26)
is correct since we focus on the $(1/4)-$sub-Gaussian case. **11.** The factor of $2^{-5}$ in the definition of $\Delta$ on line 544
is to make sure $\sqrt{2\Delta B/K} \leq 1/4$ in Eq.(31) so that we can lower bound the averaged regret. **12.** Note that $\beta$ in the
proof of Thm. 3 is really just a symbol, and one could replace $\beta$ with $\theta(0)$. Another way to understand the proof of
Thm. 3 is as following: for *any* Pareto optimal rate $\theta$, it satisfies the lower bound in Eq.(35); meanwhile, the rate on
the RHS of Eq.(35) is achieved by Alg. 1 with input $\beta = \theta(0)$. Alg. 1 is thus Pareto optimal.

**Review 4: 1.** Although Alg. 1 uses MOSS (explained in detail in [2, 14]) as a subroutine, it is *very* different from simply
fine-tuning MOSS , which fails arbitrarily when $n$ is large, or applying MOSS on a subset, which will not lead to Pareto
optimal algorithms, as discussed in Remark 2. The innovative core of Alg. 1 lies in summarizing information obtained in
iteration $i$ as a virtual arm $\widetilde{\nu}_i$. **2.** The setting with $(1/4)-$sub-Gaussian is only for convenience in calculations and could
be generalized to the $\sigma^2-$sub-Gaussian case, for any $\sigma$. **3.** Eq. after line 115 defines the hardness level of a given problem,
and Eq. after line 120 classifies problems in terms of their hardness levels. **4.** Alg. 1 is different from the Distilled
Sensing by Haupt et al 2009 since the latter only applies to very special sparse settings where optimal arms are those
with non-zero means and all other arms have zero means. **5.** Our algorithms achieves the state-of-the-art performance in
adapting to *unknown* $\alpha$. **6.** RestartingEmp (on line 256-259) represents the empirical version of Alg. 1 by allowing the
reuse of statistics. Note that we are also comparing to an algorithm, i.e., QRM2, that allows the reuse of statistics [12].

**Review 5: 1.** Our setting could be generalized to the case with multiple $\epsilon$-good arms without modification in algorithms
and (as long as $\epsilon \leq 1/\sqrt{T}$) the theoretical results hold up to negligible factors (see line 98-103; $\epsilon \leq 1/\sqrt{T} \Rightarrow \epsilon T \leq$
$\sqrt{T}$). **2.** The lower bound in Section 2 is in the minimax sense, so it suffices to reduce to the single-best arm case. A
lower bound of the order $\Omega(\sqrt{T(n-m)/m})$ ($\approx \Omega(T^{(1+\alpha)/2})$ as long as $T^\alpha \geq 2$) for the $m$-best arms case could be
obtained following similar analysis in Chapter 15 of [21]. **3.** Our results in Thm. 3 show that, in the minimax sense over
$\mathcal{H}_T(\alpha)$, suffering a rate of 1 over a certain range of $\alpha$ is *unavoidable* for algorithms on the Pareto frontier. Better bounds
might be obtained when restricting ourselves on a subset of $\mathcal{H}_T(\alpha)$, but not in general. **4.** When prior knowledge on $\alpha$
is unavailable, we recommend setting $\beta = 0.5$ and applying RestartingEmp in practice since it achieves performance
very close to the oracle algorithm with *known* hardness level. Increasing $\beta$ provides worse performance on small $\alpha$ but
better performance on larger $\alpha$. **5.** The setting with knowledge of the value $\mu_\star$ was previously studied in [23]. Besides,
we allow mis-specification in $\mu_\star$ (see point 2 in response to Reviewer 1). **6.** Similar experimental results are obtained
after averaging over 500 trials. $T = 50000$ is intentionally chosen to create the tension between $n, m$ and $T$. **7.** The
algorithm is *valid* in the sense that it selects an arm $A_t$ for any $t \in [T]$, i.e., it does not terminate before time $T$.

[Meta-Review · NeurIPS 2020]

This submission considers scenarios is which the number of arms is large (compared to the horizon) and there are multiple best arms. There has been many technical discussions of several possible limitations of the current work but the reviewers agreed that the contribution was truly novel and inspiring. The post discussion consensus was higher due to the relevant answers of the authors, with a general agreement that the submission deserves to be accepted.